# A low-complexity linker as a driver of intra- and intermolecular interactions in DNAJB chaperones

Billy Hobbs [1], Noor Limmer[1], Felipe Ossa[1], Ella Knüpling [1,4], Samuel Lenton[2], Vito Foderà [2], Arnout P. Kalverda[3] & Theodoros K. Karamanos [1]✉

J-domain proteins (JDPs) act as major regulators of the proteostasis network by driving the specificity of the Hsp70 machine. Their important functions are mediated by a low-complexity glycine-/phenylalanine-rich region (GF-linker) that links the folded J-domain with the substrate binding domain. Recently, we and others have shown that in an autoinhibited JDP state, an α-helix formed within the GF-linker blocks the Hsp70 binding site on the J-domain. However, the role of the disordered GF-linker in autoinhibition and how the latter is released, are still not understood. Here, using autoinhibited DNAJB1 and DNAJB6 constructs, we show that in combination with the J-domain, the GF-linker creates a hydrophobic, partially collapsed cluster that shows a remarkable degree of long-range structural communication, disruption of which can lead to destabilisation of autoinhibition. Apart from this crucial intramolecular role, we reveal that the GF-linker can also be recognised by the substrate-binding domain of Hsp70 and dictate the lifetime of the entire JDP–Hsp70 complex. Strikingly, the GF-linkers of DNAJB1 and DNAJB6 display distinct structural properties that lead to different Hsp70 binding kinetics, showing that the behaviour of the GF-linker can vary dramatically even within the same class of JDPs.

Proteins must navigate a complex conformational energy landscape to achieve their functional form. This landscape includes partially folded or misfolded states that are potentially toxic or lead to loss of protein function and thus need to be tightly regulated to ensure cell viability[1]. Protein maintenance is performed by the proteostasis network in which molecular chaperones are the key players[2–4]. One of the most critical components of the chaperone network is the Hsp70/JDP (J-domain protein, also known as Hsp40) system, which is involved in a multitude of housekeeping functions[5,6] and is also implicated in numerous diseases[7,8]. The human Hsp70 family contains 13 members, including Hsc70, the constitutively expressed cytosolic isoform, which is the protein used in this study. Using their highly dynamic structures, JDPs deliver specific substrates to Hsp70 and thus act as crucial drivers of the specificity of the powerful Hsp70[9].

A and B class JDPs comprise an N-terminal J-domain which binds the ATPase domain of Hsp70, followed by a glycine/phenylalanine-rich region (GF) and variable C-terminal substrate binding domains (CTDs)[10]. Despite many studies demonstrating the importance of the GF-linker[11,12], the details of how it regulates Hsp70 binding and eventually chaperone activity are only now beginning to unravel. Using DNAJB6, a class B JDP known for its anti-aggregation function[13–15], we have previously shown that residues 96 to 104 form a helix (helix 5) that blocks the binding of the J-domain to Hsp70 (helices 2 and 3)[16], creating a closed/autoinhibited state[17]. Autoinhibited states for other DNAJBs, including DNAJB1[18] and yeast Sis1[19], have since been

[1]Department of Life Sciences, Faculty of Natural Sciences, Imperial College London, London, UK. [2]Department of Pharmacy, University of Copenhagen, Copenhagen, Denmark. [3]Astbury Centre for Structural Molecular Biology, University of Leeds, Leeds, UK. [4]Present address: The Francis Crick Institute, London, UK. ✉e-mail: t.karamanos@imperial.ac.uk

discovered, suggesting that autoinhibition might be common in DNAJBs. Importantly, class A JDPs (DNAJAs), even though they seem to contain helix 5 in their sequences, are not autoinhibited[18], indicating that other factors also contribute to autoinhibition and the functional diversity of the various JDP classes. Further highlighting the importance of autoinhibition in DNAJB6, various mutations within helix 5 and its interface with the J-domain have been linked with the autosomal muscle disorder limb-girdle muscular dystrophy type 1D (LGMDD1)[20–22] in humans by promoting the formation of the open/uninhibited DNAJB6 state[23].

Apart from helix 5, the GF-linker can be further divided into an N-terminal G-rich region and a C-terminal peptide that is enriched in aromatic residues (phenylalanine and tyrosine) that appear in pairs with a small hydrophilic residue (Fx-repeats, Fig. 1A, B). Interestingly, the Fx-repeats are absent from DNAJAs, while the length and aromatic content of the Fx-repeats region differs even within DNAJBs (Fig. 1A). It is thus intriguing to hypothesise that these small changes in the composition of the Fx-repeats play a role in differentiating the structure and chaperoning functions of specific DNAJB isoforms. Indeed, the GF-linker in the context of DNAJB1, despite being less hydrophobic (containing only three Fx-repeats), is more rigid and comprises an extra helical region (termed αL, Fig. 1C) in comparison to the GF-linker of DNAJB6 (Fig. 1D), which includes four Fx-repeats. Even though both regions largely lack defined structure, mutations in the G-rich region cause Hsp70-dependent cell death in yeast[24], while in humans, Phe to Leu substitutions in the Fx-repeats region of DNAJB6 cause the most severe LGMDD1 phenotypes[25]. Due to their increased dynamics, the impact of these two regions in autoinhibition and its release, and any potential contribution to Hsp70 binding has yet to be structurally characterised.

Here, we use solution NMR in combination with small-angle X-ray scattering (SAXS) in order to elucidate the role of the G-rich and Fx-repeat regions in the correct docking of helix 5 in both DNAJB6 and DNAJB1. We show that the GF-linker of DNAJB6, but not that of DNAJB1, uses its Fx-repeats to transiently collapse onto the J-domain even in the absence of helix 5. Using a panel of aromatic-to-aliphatic substitutions, we reveal a remarkable degree of long-range structural communication between the G-rich region, Fx-repeats and helix 5 which leads to rigidification of the GF-linker in DNAJB6 and undocking of helix 5 in DNAJB1. The striking differences in the conformational properties of the two DNAJBs suggests that the GF-linker is critical for the stability of helix 5 and plays a class-dependent-role in autoinhibition release. Finally, using relaxation-based NMR methods, we show that the Fx-repeats can be specifically recognised by the substrate-binding domain (SBD) of Hsc70 demonstrating that the GF-linker also plays a crucial role in intermolecular interactions. Notably, the differences in GF-linker conformation between DNAJB1 and DNAJB6 translate into distinct interaction modes with Hsc70, suggesting that GF-linker dynamics influence Hsp70 binding affinity and kinetics. These results provide insights into the critical role of the disordered GF-linker in both intra- and inter-molecular interactions in DNAJBs, hinting at a potential Hsp70-dependent autoinhibition release mechanism. Extending further than JDPs, our findings have implications about the importance of low complexity linkers in mediating the cooperation of intrinsically disordered proteins/regions and folded domains.

## Results

### Long-range interactions between the J-domain and GF-linker in the autoinhibited state

To investigate the role of the GF-linker in autoinhibition, we first performed a detailed NOE analysis on a uniformly $^{13}$C, $^{15}$N-labelled DNAJB6 construct that contained JD, GF and helix 5 (JD-GF-α5, Fig. 1B, D) using a set of aliphatic-aliphatic and aromatic-aliphatic $^1$H-$^1$H NOEs (Fig. 1E, F and Supplementary Fig. 1). Such analysis was not possible in earlier studies of DNAJB6[17] as the samples used to solve its solution NMR

structure were specifically methyl-protonated, but otherwise highly deuterated and therefore only a minimal set of NOEs could be collected. Various long-range NOEs were observed between the N-terminal residues of the G-rich region and helix 3, including the sidechains of L73 and V55 (Fig. 1E, F and Supplementary Fig. 1). Importantly, the same residues show NOEs to the Fx-repeats (Fig. 1E, F), implying various hydrophobic contacts between the GF-linker and the J-domain in the autoinhibited state of DNAJB6, an observation that is also evident in the structure of DNAJB1 (Fig. 1C). Unfortunately, a more detailed interpretation of the observed aromatic NOEs in terms of distances was not possible due to the highly overlapped Phe region of the aromatic HSQC spectrum of DNAJB6 JD-GF-α5 which also suffers from strong-coupling artefacts for residues in the Fx-repeats (Supplementary Fig. 1A). Despite the various NOEs shown in Fig. 1E, the GF-linker in DNAJB6 JD-GF-α5 retains a high degree of dynamics as evidenced by an average hetNOE value of <0.5 (600 MHz)[17] showing that it can adopt multiple, 'non-specific' conformations.

### Residual structure in the GF-linker enables its collapse onto the J-domain

The NOE analysis presented in Fig. 1E indicates that, in the closed/autoinhibited DNAJB6 state, the GF-linker, including the Fx-repeats, makes contacts with the J-domain. These contacts may either result from the docking of helix 5 onto the J-domain or they could drive the correct packing of helix 5 with helices 2 and 3. To differentiate between these scenarios, we used two DNAJB1 and DNAJB6 constructs truncated immediately after the J-domain (JD, Fig. 1B), or just prior to helix 5 (JD-GF, Fig. 1B). Large chemical shift differences between JD-GF-α5 and JD-GF are observed for both proteins, consistent with undocking of helix 5 (Supplementary Fig. 2A, B). To control for the inevitable, non-specific effect that the addition of any 27-residue-long polypeptide would have on the resonances of the J-domain, we swapped the GF-linker with a polypeptide of the same length consisting of 9 Gly-Ser-Ser repeats (termed JD-GSS) that has been shown to behave as a true random coil[26]. Thus, comparing the spectra of the J-domain in the contexts of JD-GF and JD-GSS allows one to discriminate between the contacts the native linker forms with the J-domain in the absence of helix 5, versus the non-specific contacts of a fully disordered linker (Supplementary Fig. 2C–E). In the case of DNAJB1, we find very few differences between the JD-GF and JD-GSS spectra (Supplementary Fig. 2E) consistent with the native GF-linker behaving similarly to a random coil linker when helix 5 is not present. On the other hand, in DNAJB6, J-domain resonances in helices 1, 3, and 4 are significantly more perturbed in the presence of the GF versus the disordered GSS (Fig. 2A), showing that the native linker makes contacts with the J-domain even in the absence of helix 5.

Focusing on DNAJB6, we set out to probe the conformation of its GF-linker that potentially enables the interactions with the J-domain. To start with, we calculated the secondary structure propensities of the DNAJB6 JD-GF construct based on its assigned backbone resonances. As seen in Fig. 2B the removal of helix 5 has little effect on the helical propensity of the J-domain. However, it forces a more extended conformation in the Fx-repeats in comparison to their structure in JD-GF-α5. To investigate whether the local extension of the GF-linker has an influence on the global conformational ensemble of the JD-GF of DNAJB6, SAXS was used to determine its compaction and flexibility. As expected, the Kratky plot for JD-GF shows that this construct is more flexible than the globular JD or the closed/autoinhibited JD-GF-α5 (Fig. 2C). Moreover, the radius of gyration ($R_g$) for JD-GF was calculated at 17.3 ± 0.2 Å, a value larger than that of JD (13.8 ± 0.1 Å) and JD-GF-α5 (15.7 ± 0.1 Å), but significantly lower than that of JD-GSS (20 ± 0.1 Å) or the theoretical value calculated for the J-domain coupled with a fully disordered GF-linker (~19 Å) suggesting that it is at least partially collapsed (Fig. 2C). Taken together, the chemical shift and SAXS data presented in Fig. 2 clearly indicate that in the open state of DNAJB6,

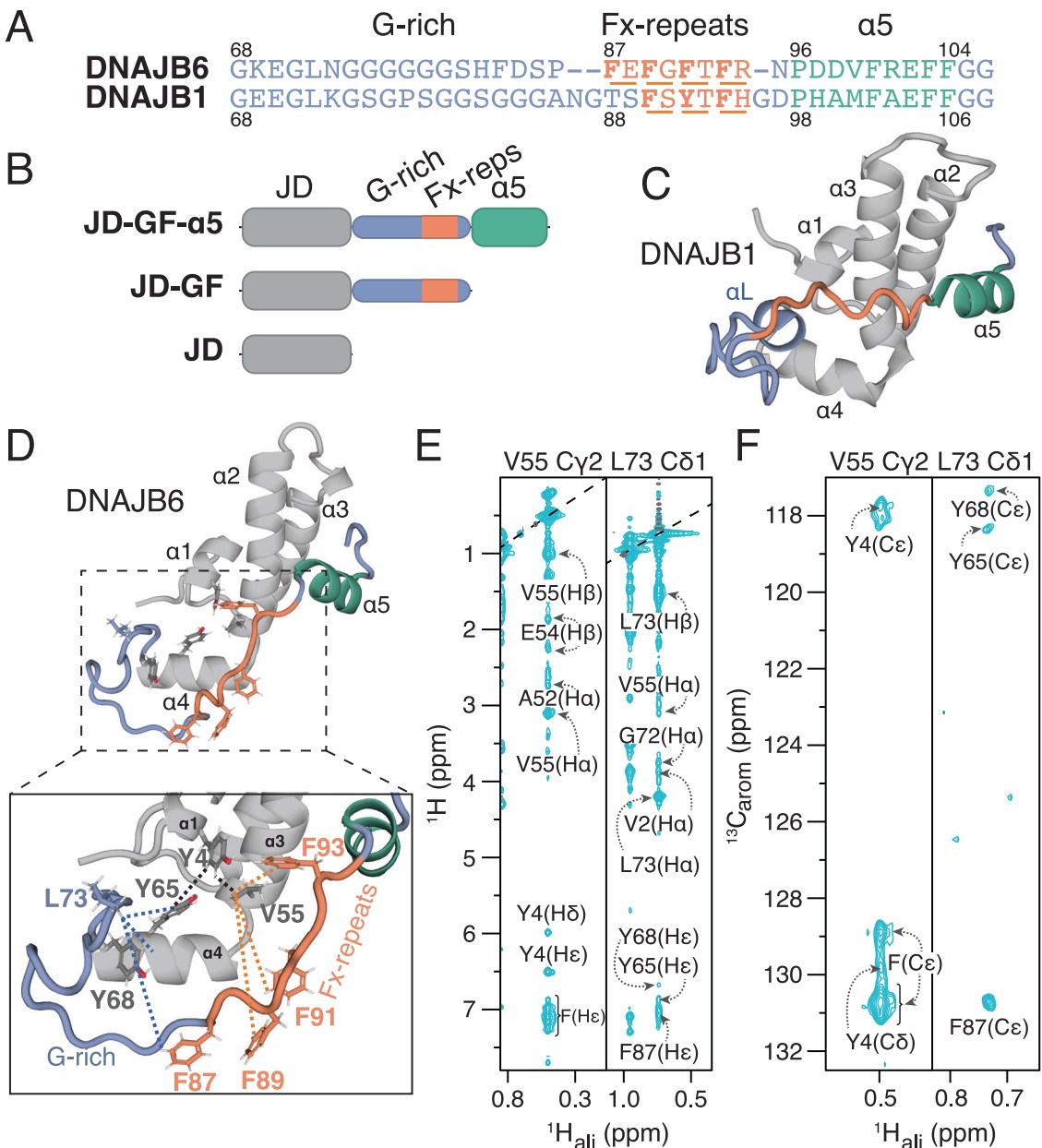

**Fig. 1 | Contacts between the GF-linker and J-domain in autoinhibited DNAJB6.**
**A** Sequence alignment of DNAJB6 and DNAJB1 GF-linkers highlighting the G-rich region, Fx-repeats and helix 5. **B** Domain architecture of the constructs used in this work using the same colour-code are as in (**A**). **C** Solution NMR structure of DNAJB1 (PDB: 6Z5N). **D** Solution NMR structure of DNAJB6 (PDB: 6U3R) with key J-domain and GF-linker residues highlighted in an inset. The G-rich region is shown in light blue, the Fx-repeats in orange, helix 5 in turquoise and the J-domain in grey. **E** Strips from 3D aliphatic NOESY-HMQC and (**F**) 3D aromatic HMQC-NOESY-HMQC spectra. The data were collected on a 1 mM $^{13}$C,$^{15}$N-labelled DNAJB6-GF-α5 sample in 100% D$_2$O at 25 °C at 600 and 800 MHz, respectively.

residual structure in the GF-linker enables its (partial) collapse onto the J-domain. Interestingly, this observation is not true for DNAJB1, showing that the small differences in the amino-acid composition of the GF-linker within DNAJBs lead to different GF-linker conformational ensembles.

**Hydrophobic contacts promote linker packing**
Having established the local and global conformational propensities of DNAJB6's GF-linker, its dynamics in the autoinhibited (JD-GF-α5) and open (JD-GF) states were assessed by NMR relaxation methods (Fig. 3A–C). $^{15}$N-$R_1$ rates for both constructs show little variation in the J-domain, with hetNOE values measured at ~0.8 (600 MHz) consistent with a folded domain of 7 kDa. On the other hand, $^{15}$N-$R_2$ rates are generally lower for the α1-α2 loop and helix 4, resulting from a slightly

anisotropic J-domain diffusion tensor (D$_{||/\perp}$ ~ 1.5 for JD alone, Supplementary Fig. 3A) in which these structural elements are positioned perpendicular to the main axis. Reflecting the semi-globular nature of JD-GF-α5, its anisotropy reduces slightly to 1.2, while that of JD-GF is ~1.6, similar to the one observed for isolated JD. $^{15}$N-$R_2$ rates are also lower for residues in the α2-α3 loop indicative of a higher degree of ps-ns dynamics for this region which lacks defined secondary structure.

As expected, large differences are observed for GF-linker residues between autoinhibited JD-GF-α5 and open JD-GF. In the autoinhibited state, $^{15}$N-$R_2$ and hetNOE rates in the GF-linker drop sharply after residue 69 and increase gradually after residue 80 to reach a plateau in helix 5 (Fig. 3B, C). Surprisingly, in the context of the open JD-GF, residues 70–74 show $^{15}$N-$R_2$ rates similar to those of residues in helix 4, while their hetNOE values are substantially increased to ~0.7 from ~0.5

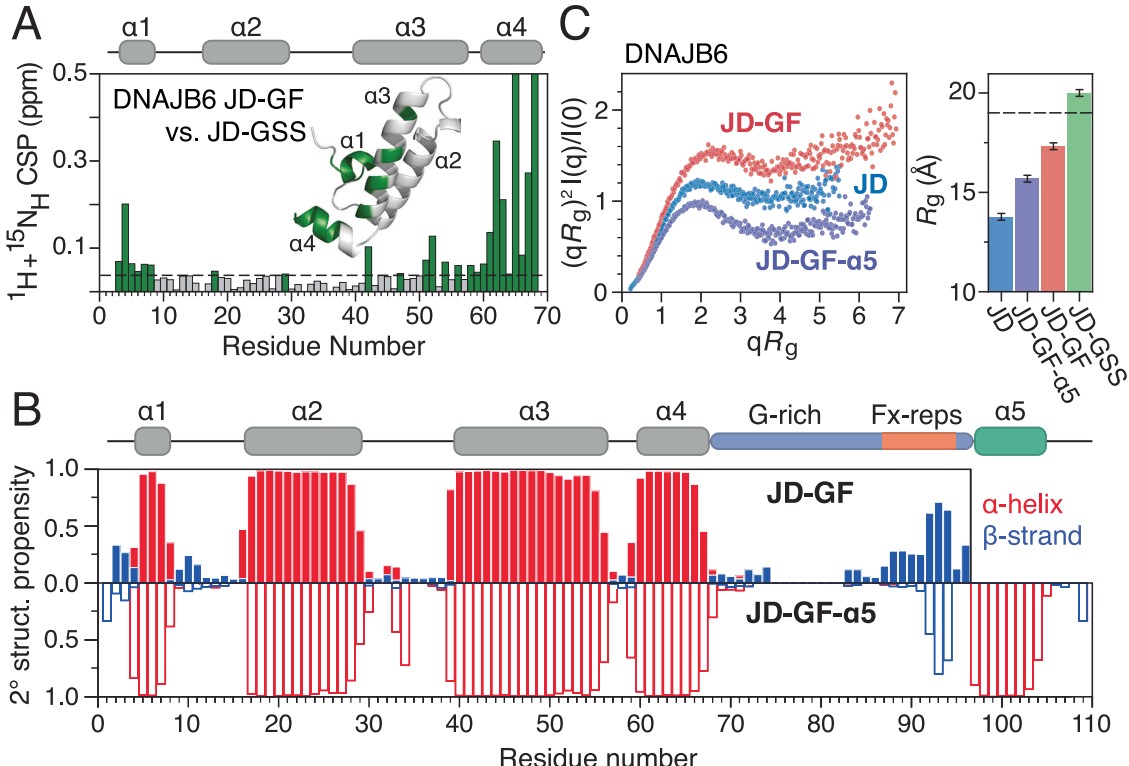

**Fig. 2 | Residual secondary and tertiary structure in DNAJB6 JD-GF. A** Combined $^1$H, $^{15}$N chemical shift perturbations between DNAJB6 JD-GF and JD-GSS constructs. Dashed line shows 2 corrected standard deviations, residues with CSPs above this cutoff (see Methods) are coloured green and are highlighted on the structure of JD-GF (inset). **B** Secondary structure propensities derived from the assigned backbone chemical shifts ($^{13}$Cα, $^{13}$Cβ, $^{13}$C′, $^{15}$N and $^1$H$_N$) for DNAJB6 JD-GF (top, solid bars) and JD-GF-α5 (bottom, open bars). Propensities for α-helices and β-strands are in red and blue, respectively. **C** Kratky plots (left) and $R_g$ values (right) for DNAJB6 constructs. The dashed line represents a theoretical $R_g$ value of JD-GF, using an ensemble of 1000 structures in which the GF-linker is completely disordered. Error bars are calculated from the Guinier fit.

in the autoinhibited JD-GF-α5, suggesting that the N-terminal portion of the G-rich region gets more ordered in the absence of helix 5. Regarding the Fx-repeats, a characteristic up/down pattern is observed in the hetNOE data, with the highest values of ~0.25 (600 MHz) corresponding to Phe87, Phe89 and Phe91 indicating reduced fast time-scale motions for these residues (Fig. 3C).

Guided by the relaxation data and despite the highly flexible nature of the JD-GF construct (Fig. 2C) an NOE analysis was conducted on the open DNAJB6 JD-GF. Although no long-range $^1$H-$^1$H NOEs were observed involving residues in the Fx-repeats, strong NOEs were observed between Leu73 and hydrophobic residues in the N-terminus and helices 3 and 4 (including Tyr4, Tyr5, Val55, Ile64, Tyr65, Tyr68, see Fig. 3D) showing that a hydrophobic cluster is formed between the G-rich region and the side of the J-domain facing helix 4. While similar contacts were observed in the autoinhibited state (Fig. 1E), these seem to be stronger in the open JD-GF as judged by the intensity of the NOE cross-peaks (Supplementary Fig. 3B). This observation rationalises the reduced motions in the G-rich region deduced from the relaxation data shown in Fig. 3A–C and suggests that transient interactions with this partially exposed hydrophobic cluster may underlie the reduced dynamics of the Fx-repeats.

**Long-range allosteric communication in the GF-linker of DNAJB6**

So far, we have shown that the GF-linker uses its hydrophobic residues to interact with the J-domain in both the autoinhibited/closed and open states of DNAJB6. However, due to signal overlap of the resonances belonging to the residues in the Fx-repeats (Supplementary Fig. 1A), the role of specific aromatic residues is difficult to establish. To

overcome this issue, a panel of single point mutants was used to dissect the effect of substituting individual aromatic residues to leucine in the Fx-repeats of the autoinhibited JD-GF-α5. Such mutations (i.e. F91L and F93L) in DNAJB6 cause the earliest onset and most severe LGMDD1 phenotypes[25] despite only minimally increasing the population of the open state[23]. Consistent with previous results[23], the F91L substitution had a profound effect on the spectrum of DNAJB6 with the observed chemical shift perturbations being highly localised around the site of the mutation and residues 70–74 in the G-rich region (Fig. 4A). Importantly, no chemical shift differences are observed in helix 5 and its interface with the J-domain, clearly indicating that the underlying structural changes that give rise to the large chemical shift perturbations of Fig. 4A are not caused by helix 5 undocking. Instead, the Phe to Leu substitution at position 91 causes an allosteric reorganisation in the G-rich region, with residues 70–74 folding into a stable helix as revealed by their backbone chemical shifts (Fig. 4B). Another prominent LGMDD1 mutation, F93L, behaves similarly, although the helical propensity of the G-rich region is about 50% (Supplementary Fig. 4). Interestingly, a small helical element in the N-terminus of the G-rich region is a common feature in JDPs, including DNAJB1 (αL in Fig. 1C) and Sis1. It is therefore interesting to hypothesise that removing an aromatic residue from the GF-linker of DNAJB6 induces a 'DNAJB1-like' structure in which αL is present. To further validate this hypothesis, one-bond $^1D_{NH}$ and two-bond $^2D_{C'H}$ residual dipolar couplings (RDCs) were measured on the F91L mutant as described in Robertson et al. (Supplementary Fig. 5A)[27]. Excellent fits were produced when fitting the RDCs using a singular value decomposition approach[28] to an AlphaFold model of autoinhibited DNAJB6 JD-GF-α5 in which residues 70–74 are in a αL conformation (Supplementary Fig. 5B) with an

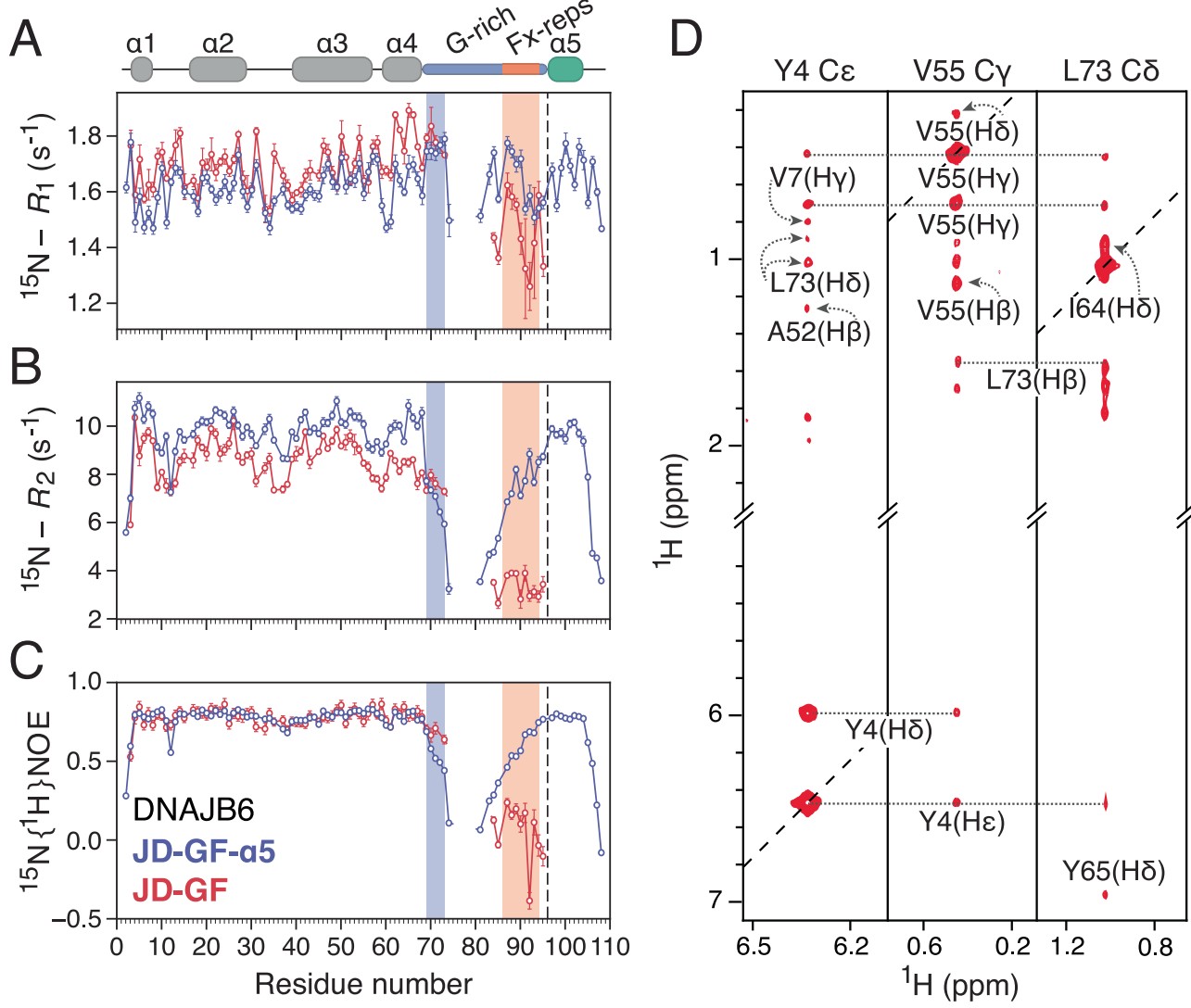

**Fig. 3 | Linker dynamics in the autoinhibited and open DNAJB6 states.** Comparison of relaxation data for JD-GF (red) and JD-GF-α5 (blue): **A** $^{15}$N-$R_1$, **B** $^{15}$N-$R_2$ and (**C**) $^{15}$N {$^1$H} heteronuclear NOE. The $^{15}$N-$R_2$ values were calculated from $^{15}$N-$R_{1\rho}$ values recorded with a 1.5 kHz spin lock field after correction for $^{15}$N-$R_1$. Data were recorded at 600 MHz and 25 °C on a 200 µM sample of $^{15}$N-labelled JD-GF-α5 and a 185 µM sample of $^{13}$C, $^{15}$N-labelled JD-GF. Missing datapoints in the GF-linker correspond to overlapping Gly residues and the vertical dashed line represents the last residue of the JD-GF construct. Error bars represent the errors calculated based on the covariance matrix of a single exponential fit. **D** Strips from 3D aromatic (left most panel) and 3D aliphatic (middle and right panels) NOESY-HMQC spectra recorded on 300 µM $^{13}$C, $^{15}$N-labelled DNAJB6 JD-GF in 100% D$_2$O at 25 °C, 700 MHz.

$R_{factor}$[29] of 42% and 41% for $^1D_{NH}$ and $^2D_{C'H}$ respectively (Fig. 4C, D). Since undocking of helix 5 would significantly alter its orientation with respect to the J-domain and in turn its RDC values, the RDC analysis presented in Fig. 4 consolidates the conclusion that the Phe to Leu mutations in the GF-linker of DNAJB6 do not affect the docking of helix 5. Moreover, it is also evident that the orientation of the new helix involving residues 70–74 is compatible with that of αL in DNAJB1. Overall, by taking advantage of the sensitivity of RDCs to structural changes, we show that removing an aromatic residue from the Fx-repeats of DNAJB6 causes a long-range reorganisation in the GF-linker which now adopts a more rigid, DNAJB1-like structure with helix 5 still docked.

### The role of the GF-linker in releasing autoinhibition

Given the different conformational properties of the autoinhibited DNAJB1 GF-linker in comparison to that of DNAJB6 (Fig. 1C, D) we set out to investigate the role of specific aromatic residues in the less hydrophobic DNAJB1 linker (Fig. 1A). Towards this direction, DNAJB1

residues Y92 and F94, equivalent to DNAJB6's F91 and F93, respectively (Fig. 1A), were mutated to leucine in the JD-GF-α5 construct. In contrast to the highly localised effect of F91L in DNAJB6 (Fig. 4A), the equivalent Y92L mutation in DNAJB1 JD-GF-α5 caused widespread chemical shift perturbations that extend to residues in helix 5 and helices 3 and (to a lesser degree) 2 that are in the interface of helix 5 with the J-domain (Fig. 5A). Similar observations were made for DNAJB1 F94L (Supplementary Fig. 6A). Interestingly, the chemical shifts of residues in helices 2 and 3 in both DNAJB1 Y92L and F94L JD-GF-α5 move towards their positions in the open JD (Fig. 5B), suggesting that these Phe to Leu mutations cause a transition from the autoinhibited to the open state. The exchange lies in the fast regime of the chemical shift timescale, allowing the populations of the open state to be calculated from the positions of the Y92L and F94L JD-GF-α5 resonances at ~40% and ~70%, respectively (Fig. 5B). To confirm that the observed chemical shift perturbations are due to undocking of helix 5, and not the outcome of other structural rearrangements within the GF-linker as observed in DNAJB6 (Fig. 4), the $^{15}$N-$R_2$ rates of Y92L and F94L DNAJB1 JD-GF-α5

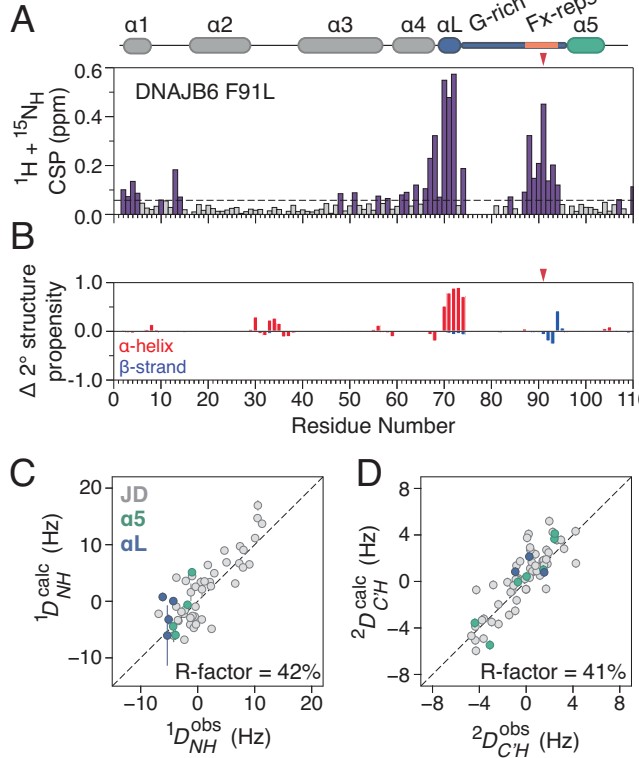

**Fig. 4 | Effect of phenylalanine substitutions on the structure of DNAJB6.**
**A** Combined $^1$H, $^{15}$N chemical shift perturbations between wild-type DNAJB6 JD-GF-α5 and F91L constructs. Dashed line shows 2 corrected standard deviations, residues with CSPs above this cutoff (see Methods) are coloured purple. **B** Difference in secondary structure propensities between WT DNAJB6 JD-GF-α5 and F91L using values calculated based on the assigned backbone chemical shifts ($^{13}$C$_α$, $^{13}$C$_β$, $^{13}$C′, $^{15}$N and $^1$H$_N$). The wild-type propensities were subtracted from those of F91L, and therefore positive values correspond to increased secondary structure propensity in the mutant. The location of the F91L mutation is denoted by a red arrow. Agreement between measured (**C**) $^1D_{NH}$ and (**D**) $^2D_{C'H}$ with those back-calculated from an AlphaFold model of DNAJB6 JD-GF-α5 in which residues 70–74 are in a helical conformation (Fig. 1C and Supplementary Fig. 5B). JD, helix 5 and residues 70–74 are shown in grey, green and purple respectively. Error bars were estimated based on the signal to noise ratio of each resonance and the R-factor was calculated using the method of Clore and Garrett[29]. RDC data were collected on 150 µM $^{13}$C,$^{15}$N-labelled samples of F91L DNAJB6 JD-GF-α5 at 600 MHz and 25 °C with and without 13 mg/mL Pf1 bacteriophage as the alignment medium.

(Fig. 5C) were measured. Elevated $^{15}$N-$R_2$ values, characteristic of chemical exchange contributions to relaxation ($R_{ex}$), were observed for residues A100, A103 and F106 in helix 5 in Y92L in comparison to wild-type DNAJB1 JD-GF-α5, with their values being further increased in F94L (Fig. 5C). The F94L cross-correlated $η_{xy}$ rates which are immune to exchange, are almost uniform across helices 1–5 (Supplementary Fig. 6B), confirming the presence of $R_{ex}$ terms for helix 5 residues in the mutants. Owing to the large difference in the $^{15}$N chemical shifts between the open and closed/autoinhibited states for R26 and L29 in helix 2, these residues also show small $R_{ex}$ contributions (Fig. 5C) which can be used alongside the known populations and chemical shifts of the open state to place the exchange rate ($k_{ex}$) of the open to close transition in the 20,000 s$^{-1}$ range. Since the $η_{xy}$ rates (which report on the overall tumbling of the molecule) of residues in helix 5 are essentially the same as those of residues in helices 1–4, it is evident that even in the 'open' state helix 5 still interacts with the J-domain. As has been observed in molecular dynamics simulations[30], these interactions seem to have been substantially weakened by the F to L mutations in the GF-linker, causing JD-GF-α5 resonances to move significantly

towards their positions in free JD as shown in Fig. 5B. If true, this scenario should be reflected in the hydrogen exchange rates of the GF-linker and helix 5, as these are sensitive to even transient exposure to solvent.

To prove this hypothesis, amide hydrogen exchange rates were measured on Y92L and F94L, and compared to wild-type DNAJB1 JD-GF-α5 using a WEX-III experiment[31], ideally suited for measuring fast hydrogen exchange as expected for the disordered GF-linker. Indeed, amides in the GF-linker of the wild-type were found to exchange with solvent in the 10–40 s$^{-1}$ range, significantly faster than the amides of the J-domain residues which exchange at ~0.5 s$^{-1}$ (Supplementary Fig. 7). In comparison to wild-type DNAJB1 JD-GF-α5, Y92L and F94L cause a 1.5–4-fold increase in the hydrogen exchange rates of amides in the G-rich region, the Fx-repeats but most importantly of those in helix 5 (Supplementary Fig. 7). Converting the exchange rates to protection factors (Fig. 5D) reveals that Y92L and F94L has caused a substantial destabilisation of helix 5 while residues in the J-domain remain largely unaffected. Overall, the chemical shift, $^{15}$N relaxation and hydrogen exchange data (Fig. 5) show that GF-linker mutations destabilise helix 5, causing it to undergo fast exchange with open conformations in the µs timescale while still making contacts with the J-domain.

Comparing data in Fig. 4 to those in Fig. 5, the effect of aromatic to leucine mutations of equivalent residues in the GF-linker of DNAJB6 (Fig. 4) and DNAJB1 (Fig. 5) differs significantly. We note that DNAJB6's GF-linker has four Fx-repeats in comparison to DNAJB1's three (Fig. 1A). Given that the single point F to L DNAJB6 variants (Fig. 4) adopt a semi-rigid 'DNAJB1-like' GF-linker structure with helix 5 stably bound to the J-domain, we wondered whether further reducing the hydrophobicity of the GF-linker of DNAJB6 will cause a destabilisation of helix 5 as seen for DNAJB1 in Fig. 5. To test this hypothesis the entire DNAJB6 GF-linker region was swapped for a highly disordered GSS linker of the same length but with helix 5 present. This DNAJB6 JD-GSS-α5 construct shows the characteristic $R_{ex}$ terms observed for the DNAJB1 mutants in helices 2 and 5 (Fig. 5C and Supplementary Fig. 8A) and even a reduced helical propensity in helix 5 (Supplementary Fig. 8B), indicating a destabilised autoinhibited state. Taken together, our data reveal a direct link between destabilisation of autoinhibition and reduction of the aromatic/hydrophobic content of the partially collapsed GF-linker which has been shown to correlate with the ability of DNAJBs to enhance the ATPase activity of Hsc70[23].

### The GF-linker as an Hsc70 substrate
Two binding interfaces between JDPs and Hsp70s have been well established in the literature. These include: 1) the binding of the J-domain to the groove formed between the nucleotide-binding domain (NBD) and substrate-binding domain (SBD) of Hsp70[16] and 2) the interaction of Hsp70's EEVD tail with the JDP's CTD[32]. Using evidence from kinetic and/or competition experiments, JDPs have been proposed to act as substrates for Hsp70[33,34], suggesting the presence of additional binding sites, although direct evidence for this interaction remains elusive. To investigate any potential Hsp70 binding interfaces, the interaction of DNAJB6 and DNAJB1 JD-GF with the constitutively expressed form of human Hsp70 (Hsc70) in the ATP bound state was studied by NMR. The open JD-GF constructs were used in these experiments in order to overcome the reduced affinity of autoinhibited JDPs for Hsc70 in the presence of helix 5, and also to avoid any potential long-range structural effects related with release of autoinhibition. Addition of unlabelled Hsc70 to $^{15}$N-labelled DNAJB6 JD-GF caused a global decrease in the intensities of the latter, accompanied by very small $^{15}$N exchange induced shifts ($^{15}$N-$δ_{ex}$) characteristic of exchange between the unbound species and the large(r) molecular weight complex (Supplementary Fig. 9). Indeed, the interaction of DNAJB constructs with full-length Hsc70 was found to take place in the ms timescale and thus was probed by $^{15}$N Carr-Purcell-Meiboom-Gill (CPMG) relaxation dispersion experiments as shown in

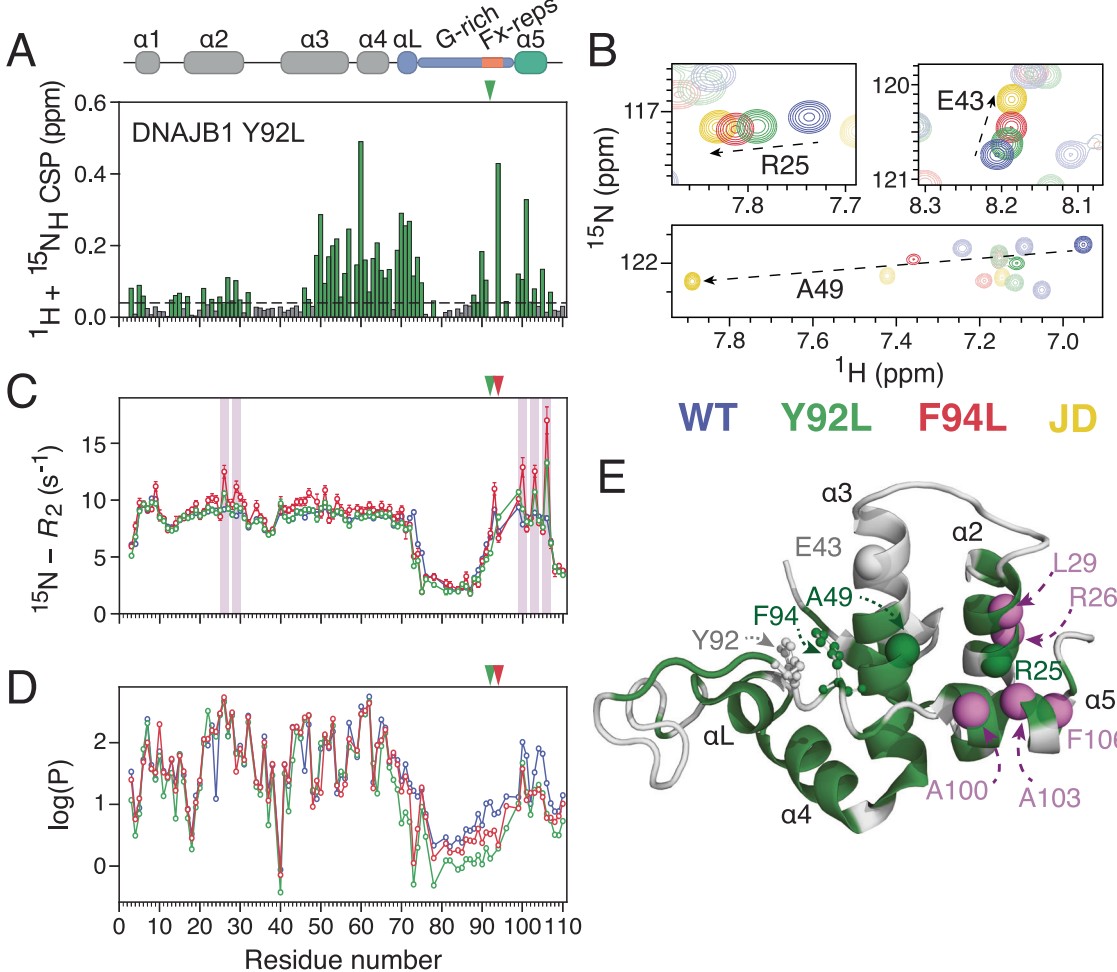

**Fig. 5 | Aromatic substitutions in the GF of DNAJB1 destabilise helix 5.**
**A** Combined $^1$H, $^{15}$N chemical shift perturbations between wild-type DNAJB1 JD-GF-α5 and Y92L constructs. Dashed line shows 2 corrected standard deviations, residues with CSPs above this cutoff (see Methods) are coloured green. **B** Regions of the $^1$H-$^{15}$N HSQC spectra of DNAJB1 JD overlayed with that of WT DNAJB1 JD-GF-α5 (blue), Y92L (green) or F94L (red). **C** $^{15}$N-$R_2$ relaxation rates for JD-GF-α5 WT, Y92L and F94L. Residues with significant $R_{ex}$ contributions to their $R_2$ rates are shaded in purple boxes. Error bars represent the errors calculated based on the covariance matrix of a single exponential fit. **D** Hydrogen exchange protection factor values for WT DNAJB1 JD-GF-α5, Y92L and F94L. The location of Y92L and F94L mutations is shown by arrows in (**A**), (**C**), and (**D**). All experiments were performed at 200 μM and 600 MHz. **E** The structure of DNAJB1 JD-GF-α5 (PDB: 6Z5N) coloured using the same colour code as in (**A**), residues shown in (**B**) are highlighted as spheres and those with significant $R_{ex}$ terms are shown as purple spheres. Tyr92 and Phe94 are shown in a stick-and-ball representation.

Fig. 6. As expected, CPMG profiles were observed for J-domain residues in helices 2 and 3, known to take part in Hsc70 binding but also for residues in the GF-linker including Phe87, Glu88, Gly90, Phe91, Phe93 and Arg94 (Supplementary Fig. 10) suggesting that the DNAJB6 linker is also involved in Hsc70 binding. Importantly, CPMG curves were flat in the absence of Hsc70 (Supplementary Fig. 11), excluding the possibility that the observed ms dynamics arise from internal DNAJB6 motions. The $^{15}$N-CPMG and $\delta_{ex}$ data fit well to a simple 2-state model which yields a $K_d$ of 380 μM for the association of DNAJB6 JD-GF with Hsc70 while $k_{ex}$ (see Methods) is ~600 s$^{-1}$ (Fig. 6A), consistent with the very small $^{15}$N-$\delta_{ex}$ values observed (Supplementary Fig. 9A).

To elucidate the role of the GF-linker in Hsc70 binding, it is instructive to compare the binding kinetics of the DNAJB6 JD-GF construct with those of the shorter DNAJB6 JD (Fig. 6B and Supplementary Table 1). The same set of J-domain residues in both DNAJB6 JD and DNAJB6 JD-GF are involved in binding, (Fig. 6B and Supplementary Fig. 12) producing an excellent correlation between their fitted chemical shifts (Supplementary Fig. 9D). However, the DNAJB6 JD–Hsc70 interaction is 2 orders of magnitude tighter with a $K_d$ of ~3 μM, indicating that even in the absence of helix 5, the partial collapse of the GF-linker onto the J-domain observed for DNAJB6 JD-GF (Figs. 2 and 3) is

enough to significantly decrease the affinity of JD to Hsc70. As the JD–Hsc70 binding interface is flat[16], the dissociation rate constant ($k_{off}$) of the DNAJB6 JD binding is ~2700 s$^{-1}$, five times faster than that of DNAJB6 JD-GF (~580 s$^{-1}$) and consistent with the 4-fold larger values of the $^{15}$N-$\delta_{ex}$ (Supplementary Fig. 9C). The significantly lower $k_{off}$ in the presence of the GF-linker strongly suggests that the linker is participating in a binding interface which dissociates much slower than the J-domain. We note here that the presence of two binding interfaces for DNAJB6 JD-GF (involving the J-domain and GF-linker respectively) could well indicate a binding reaction that is more complicated than 2-state, but in the absence of further evidence, we selected the simplest model that can describe the $^{15}$N-CPMG and $\delta_{ex}$ data. Taken together, the CPMG results clearly show that the GF-linker of DNAJB6 is able to control both the thermodynamics and kinetics of Hsc70 binding.

For DNAJB1 JD-GF on the other hand, the situation is quite different. CPMG curves were only observed for JD residues but not for those in the GF-linker (Fig. 6C and Supplementary Fig. 13) suggesting that DNAJB1's disordered GF-linker (Supplementary Fig. 2) is not involved in Hsc70 binding, which is now only mediated by JD. Moreover, the observed $^{15}$N-$\delta_{ex}$ values for the DNAJB1 JD-GF interaction with Hsc70, are of similar magnitude to those of the isolated JD

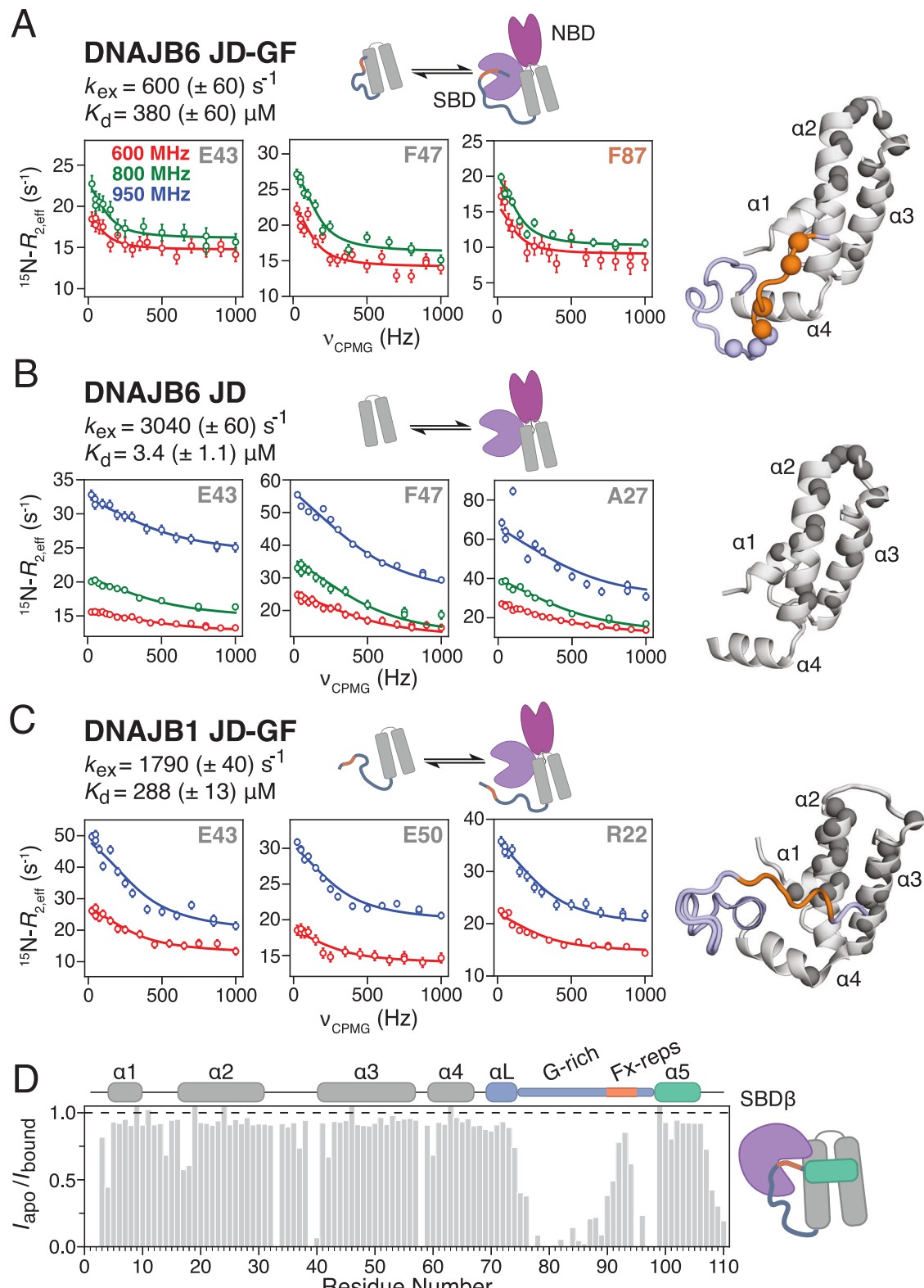

**Fig. 6 | Hsc70 binding kinetics.** Hsc70 binding to NMR visible DNAJB6 JD-GF (**A**), DNAJB6 JD (**B**) and DNAJB1 JD-GF (**C**). For each construct, the kinetic scheme used to fit the $^{15}$N-CPMG and $\delta_{ex}$ data and the obtained $k_{ex}$ (given by $k_{ex} = k_{on}^{app} + k_{off}$) and $K_d$ values are shown on the top. The raw CPMG data (open dots) are overlayed with the fitted curves at various magnetic fields as indicated on the figure. Errors bars on the CPMG data represent the standard deviation of duplicate CPMG fields. On the right, residues that show significant $R_{ex}$ curves are shown as balls on the corresponding structure (JD in grey, G-rich region in light blue and Fx-repeats in orange). All obtained kinetic parameters are shown in Supplementary Table 1. **D** Bar chart showing the ratio of the intensities of the DNAJB1 JD-GF-α5 resonances in the apo versus the SBDβ bound samples.

(Supplementary Fig. 9B, C) indicating that the DNAJB1 JD-GF binding to Hsc70 lies on the fast side of intermediate exchange on the chemical shift timescale as observed for JD alone (Fig. 6B and Supplementary Fig. 9C). Indeed, the fitted $k_{ex}$ rate for DNAJB1 JD-GF binding to Hsc70 is -1800 s$^{-1}$ closer to that of JD alone (-3000 s$^{-1}$) than that of the equivalent DNAJB6 JD-GF construct (600 s$^{-1}$). The affinity for the DNAJB1 JD-GF–Hsc70 interaction is only -300 μM, although this is due to a lower affinity of JD itself for Hsc70 in comparison to its DNAJB6 counterpart (Supplementary Fig. 14), a conclusion that is supported by preliminary CPMG analysis of the DNAJB1 JD association with Hsc70 (Supplementary Table 1).

These observations leave the question of where the GF-linker binds to Hsc70 unanswered. The SBD seems like an obvious candidate given that the GF-linker sequence of both DNAJB6 and DNAJB1 is strongly predicted to be an Hsc70 substrate[35]. This poses the further question of why only the GF-linker of DNAJB6 binds Hsc70. The small differences in the sequence composition between the GF-linker of DNAJB1 and that of DNAJB6 are unlikely to be responsible for the differences in their ability to bind Hsc70 but perhaps the compact (DNAJB6) versus the disordered (DNAJB1) nature of the two linkers might have a role in Hsc70 recognition. To resolve this apparent contradiction, the interaction of unlabelled β-stranded Hsc70 SBD subdomain (SBDβ) with DNAJB1 JD-GF-α5, a construct in which the GF-linker adopts a semi-rigid/collapsed conformation (Fig. 1C), was investigated. In this experiment, if the SBDβ was not responsible for binding the GF-linker, or if the DNAJB1 GF-linker sequence is inherently not able to bind Hsc70, no interaction would be detected. However, as shown in Fig. 6D, a drop in the intensities of residues in the GF-linker was observed in the presence of SBDβ (Fig. 6D), clearly indicating that binding is taking place in the ms timescale leading to chemical shift exchange broadening of the DNAJB1 GF-linker resonances. This result indicates that only when the GF-linker is partially collapsed as in DNAJB6 JD-GF or DNAJB1 JD-GF-α5 it can engage with SBD.

Overall, the results presented in Fig. 6 reveal differences in the way that DNAJB1 and DNAJB6 engage with Hsc70 and suggest a distinct, class-dependent ability of the GF-linker to bind Hsc70. Taken together, we were able to show that apart from playing a crucial intramolecular role by stabilising autoinhibition (Figs. 1, 2, 4 and 5) and depending on its conformational properties, the GF-linker can also act as a substrate for SBD with this interaction dictating the lifetime of the entire DNAJB–Hsc70 complex (Fig. 6).

## Discussion

JDPs play a key role in the regulation of the proteostasis network by interacting with the powerful Hsp70[10]. Despite their importance, key mechanistic details on how they drive the specificity of the Hsp70 machine remain unknown, mainly due to the high flexibility of JDPs and the transient nature of the JDP–Hsp70 interaction. The key role of the low-complexity GF-linker in controlling the ATPase activity of Hsp70 was realised early on, but a structural view of its function emerged only recently[17,18]. The discovery of the autoinhibited DNAJB state revealed an extra layer of regulation of the Hsp70 cycle through the interaction of helix 5 with the J-domain[17]. However, the disordered parts of the GF-linker are not simple bystanders in autoinhibition and its release. This is already evident by the fact that the most severe DNAJB6 LGMDD1 mutations all locate in the Fx-repeats and not within helix 5 itself[25].

Here, we show that in the autoinhibited state of DNAJB6, the GF-linker is involved in various intramolecular interactions that contribute to the docking of helix 5 (Fig. 1). Perhaps more surprisingly, using a combination of solution NMR and SAXS, we reveal that hydrophobic contacts mediated by the aromatic GF-linker residues are responsible for the collapse of DNAJB6's GF-linker onto the J-domain even in the absence of helix 5 (Figs. 2 and 3). Two key areas of long-range structural communication within the GF-linker were identified. The first one comprises the N-terminal G-rich region (residues 70–74) immediately following helix 4, which together with hydrophobic residues in the J-domain (Val55) forms a partially collapsed region on the side of helix 3. Rather remarkably, this area responds to the release of autoinhibition by becoming less dynamic. We hypothesise that this rigidification of residues 70–74 acts as a spring helping to re-establish the autoinhibited state by bringing helix 5 in close proximity to the J-domain after its undocking (Fig. 7). Interestingly, elegant biochemical and structural work on the yeast Ydj1/Sis1 has also identified residue Gly70 as being a key, Hsp70-dependent regulator of yeast proteostasis[24,36].

The second area of interest includes the Fx-repeats whose aromatic residues can maintain the GF-linker in a compact state even in the absence of helix 5, presumably facilitating its docking to the J-domain. The aromatic content of this area seems to be an important factor in determining the conformation of the GF-linker and establishing autoinhibition (Fig. 7). The four aromatic residues in DNAJB6's Fx-repeats, although still involved in long range interactions (Fig. 1) allow the GF-linker to maintain a fluid conformation. On the other

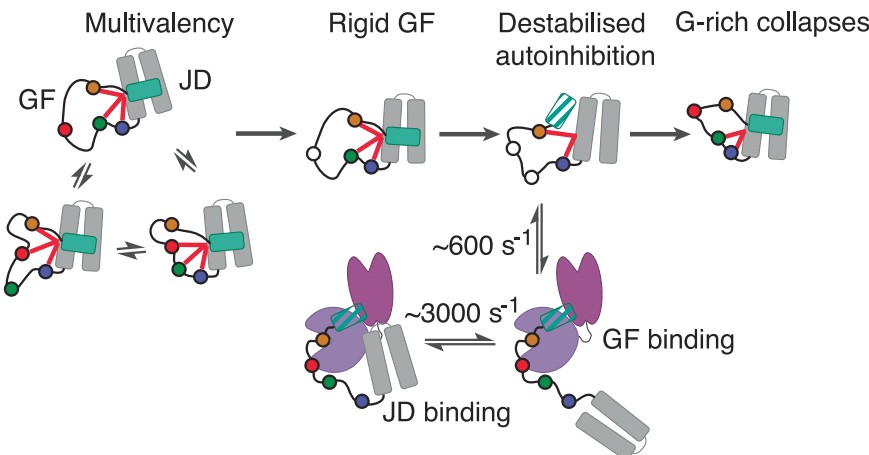

**Fig. 7 | The proposed intra- and intermolecular roles of the GF-linker.** The GF-linker can exist in various conformations including a fluid configuration (DNAJB6) stabilised by multivalent interactions (red lines) between the J-domain, G-rich region (blue spheres) and the Fx-repeats (green, red, orange spheres). Removing one of the aromatic GF-linker residues induces a rigid/specific GF-linker structure (DNAJB1, DNAJB6 F91L) which leads to destabilisation of autoinhibition if the

aromatic content of the GF-linker is reduced further. In this scenario, the G-rich region collapses against the J-domain, potentially re-establishing autoinhibition. The GF-linker is also able to bind the SBD of Hsc70 (here shown arbitrarily via the destabilised state), with this binding event being significantly slower than that mediated by the J-domain. Calculated exchange rates ($k_{ex} = k_{on}^{app} + k_{off}$) are shown for the JD and GF-linker binding.

hand, all constructs with three aromatic residues (DNAJB1 JD-GF and DNAJB6 F91L/F93L JD-GF) show a highly specific GF-linker configuration through prominent long-range interactions with residues 70–74 which adopt a helical configuration (αL, Figs. 1B, 4 and 7). Further reducing the number of aromatic residues in the Fx-repeats to two (or less), destabilises autoinhibition (Figs. 5, 7 and Supplementary Fig. 8). Interestingly, the main difference between the JD-GF of the auto-inhibited DNAJBs and that of the uninhibited DNAJAs is the complete lack of the peptide corresponding to the Fx-repeats in the latter. Given our results, we predict that DNAJB11 that localises at the endoplasmic reticulum would also not be autoinhibited due to a lack of the Fx-repeats region.

The nature of the open state bears some discussion too. A complete loss of contacts between helix 5 residues and the J-domain, as one might have expected for an 'open' state, does not take place (Fig. 5C). Instead, even in variants where the stability of helix 5 has been compromised (Supplementary Fig. 8), including those that according to their chemical shifts significantly populate the open state, helix 5 still interacts with the J-domain. This finding is in accordance with the hypotheses generated by molecular dynamics simulations[30] and suggests that extra steps need to take place in order to fully expose the Hsc70 binding site. Nevertheless, the influence of GF-linker substitutions to helix 5 is striking, as they can decrease its stability (Fig. 5C and Supplementary Fig. 7) or even cause its partial unfolding when the GF-linker is completely removed (Supplementary Fig. 8).

Apart from the crucial intramolecular roles of the GF-linker, we show that it can also dictate binding to Hsc70. The idea that JDPs can act as substrates for Hsp70 has been suggested previously for the bacterial DNAJ/DNAK system using single turnover ATP hydrolysis assays[37] and competition NMR experiments[33]. However, direct evidence for this interaction and the details of how it influences the binding of the J-domain to Hsp70, especially in light of the recently discovered autoinhibition in DNAJBs, was lacking. Here we show a remarkable ability of the low-complexity GF-linker to control the binding affinity, and kinetics of the DNAJB6–Hsc70 association. Even in the absence of helix 5, the partial collapse of DNAJB6's GF-linker onto the J-domain is enough to occlude the J-domain binding interface resulting in a two-orders of magnitude decrease in affinity for Hsc70 (Fig. 6A, B). The binding of the GF-linker to Hsc70's SBD is significantly slower than that of the J-domain to the NBD-SBD cleft and thus dictates the lifetime of the entire complex (Fig. 7). Interestingly, binding of peptide substrates to SBD is even slower and of higher affinity than that of the GF-linker binding to SBD suggesting that the substrate could still displace the GF-linker[33]. It is intriguing to hypothesise that, akin to the Phe to Leu substitutions shown in Fig. 5, GF-linker binding to SBD could destabilise the partially exposed, hydrophobic GF cluster leading to release of autoinhibition, while at the same time serving to open up the SBD for substrate binding[38]. In terms of LGMDD1, and in contrast to mutations in the helix 5 – JD interface in DNAJB6 which release autoinhibition[23], those in the GF-linker do not affect the docking of helix 5 (Fig. 4). Interestingly, all LGMDD1-associated mutations in the GF-linker of DNAJB6 (Phe89, Phe91, Phe93) localise in the Fx-repeats region that is involved in binding to SBDβ (Fig. 6). It is thus possible that the mis regulation of the proteostasis network that leads to disease is related with the GF-linker – SBDβ binding event.

Another unexpected result is the different behaviour of the DNAJB1 and DNAJB6 linkers which only minorly differ in sequence. For DNAJB1 the more-constrained GF-linker conformation in the closed/autoinhibited state (Fig. 1C) turns into a highly dynamic, almost random coil configuration when helix 5 is removed in the open state (Supplementary Fig. 2). In the case of DNAJB6 on the other hand, the dynamic nature of the linker in the closed/autoinhibited state (Fig. 1D) contrasts the collapsed configuration it adopts in the open state (Fig. 2). These different conformational properties enable the specific recognition of the GF-linker by SBDβ as shown in Fig. 6. While the disordered GF-linker of the 'open' DNAJB1 JD-GF cannot bind the SDBβ, the partially collapsed GF-linker of DNAJB6 JD-GF can. However, restricting the conformational flexibility of DNAJB1's GF-linker by introducing the 'closed' JD-GF-α5 state allows it to be recognised by the SBDβ, showcasing a conformation-specific bias for Hsc70 substrates.

More generally, the results presented demonstrate a remarkable collaboration between a low-complexity linker and an adjacent folded domain in regulating protein function. Low-complexity regions have recently attracted significant attention due to their involvement in phase separation[39,40] and nuclear transport[41]. Specifically, Phe/Gly-rich repeats in nucleoporins control selective transport through the nuclear transport complex using a network of hydrophobic interactions in a similar fashion to the way the GF-linker of DNAJBs determines binding to Hsc70. In terms of chaperones, Gly- and aromatic-rich regions are not unique to JDPs. Such regions have been shown to be important in protein refolding by the chaperonin GroEL where the highly conserved, low complexity C-terminal regions enhance chaperoning by directly interacting with the misfolding substrate[42]. In general, even though our understanding of how intrinsically disordered proteins/region function is steadily increasing, relatively little is known about how they work in conjunction with folded domains. We believe that unravelling the complex interplay between the J-domain and GF-linker is an important advance in this direction. Studies like this in the future combined with the development of force fields that are able to simultaneously deal with both folded and disordered regions[43], are likely to unlock mechanistic details on protein function that have remained elusive so far.

## Methods

### Design, expression and purification of J-domain protein (JDP) constructs

Constructs of JD alone are used as mimics of the 'open' state while JD-GF-α5 constructs have been shown to be excellent mimics of the 'closed'/autoinhibited JD state[17,18]. To isolate the impact of the disordered part of the GF-linker on the closed JD state we have removed helix 5 to create a JD-GF construct in which the conformational properties of the GF-linker and its compaction against JD can be studied without the influence of the docking of helix 5 against helices 2 and 3. All JDP constructs were produced using the protocol of Karamanos et al.[17], with some alterations. Briefly, the pET-15b plasmid containing the relevant JDP gene with an N-terminal His-tag and TEV cleavage site (synthesised by Genscript) was transformed into BL21(DE3) *E. coli* cells, grown in M9 minimal media supplemented with $^{15}NH_4Cl$ and $^{13}C$-glucose or LB at 37 °C until an $OD_{600}$ of 0.6 was reached. Expression was induced with 1 mM IPTG and allowed to continue overnight at 25 °C. The following morning, the cells were harvested for 15 min at $6200 \times g$ and 4 °C, before being resuspended in 20 mM sodium phosphate pH 8.0, 20 mM imidazole, 150 mM NaCl buffer, containing a protease inhibitor cocktail. Cells were lysed by a cell disruptor operating at 28 kPsi and the lysate was centrifuged for 40 min at $27,000 \times g$. The supernatant was loaded onto a 5 mL HisTrap HP column equilibrated in 20 mM sodium phosphate pH 8.0, 20 mM imidazole, 150 mM NaCl. A wash with 8 M urea was performed and the protein eluted with 100–200 mM imidazole. Fractions were combined and then TEV protease was added, and dialysis was carried out overnight at 4 °C against 20 mM sodium phosphate pH 8.0, 150 mM NaCl, 2 mM DTT to cleave the His-tag. The dialysate was loaded onto a HisTrap column and the flow-through was collected. The samples were then concentrated before being loaded onto a C18 reverse phase HPLC column and eluted with a gradient of acetonitrile. Fractions containing the pure protein were lyophilised and resuspended in 20 mM sodium phosphate pH 7.0, 50 mM NaCl, or 20 mM HEPES pH 7.0, 50 mM KCl, 2.5 mM $MgCl_2$ when being used for interactions with full-length Hsc70. The DNAJB6

J-domain construct with an additional C-terminal GSSC sequence for fluorescent labelling was produced as above with two alterations. The construct was produced without isotopic labelling in rich media, Terrific Broth and all purification buffers were supplemented with 2 mM DTT.

## Fluorescent labelling of DNAJB6 JD

Lyophilised DNAJB6 JD with an additional C-terminal GSSC was reconstituted into 20 mM HEPES pH 7.0, 50 mM KCl, 2 mM EDTA buffer supplemented with 5 mM DTT. Once in solution constructs were passed through a Superdex 75 Increase 10/300 GL column and exchanged into buffer without reducing agent. Labelling at the C-terminal cysteine was performed with Fluorescein-5-Maleimide (Thermo Scientific) using a 5-fold molar excess of label at 4 °C overnight, according to the manufacturer's instructions. Labelled protein was dialysed extensively, concentrated and subjected to a further round of size exclusion chromatography to ensure removal of free label.

## Expression and purification of Hsc70 constructs

Hsc70-SBDβ was purified as above until the lyophilisation stage. After lyophilisation, the protein powder was resuspended in 8 M urea to a concentration below 50 μM. The urea was then slowly removed through dialysis against 20 mM sodium phosphate pH 7.0, 50 mM NaCl.

Full-length Hsc70-T204A (ATPase deficient mutant) was fused to a TwinStrep tag with a TEV cleavage site in a pET-15b vector. Protein expression was carried out at 25 °C overnight. The following morning, the cells were harvested for 15 minutes at 6200 × *g* and 4 °C, before being resuspended in 50 mM HEPES pH 8.0, 500 mM KCl, 2 mM EDTA, 1 mM DTT containing a protease inhibitor cocktail. Cells were lysed by a cell disruptor operating at 28 kPsi and the lysate was centrifuged for 40 min at 27,000 × *g*. The supernatant was loaded onto a StrepTactin 4Flow high-capacity column, washed extensively with buffer and with 1 M KCl. Protein was eluted with 50 mM biotin. TEV protease was added to the eluent before being dialysed against 20 mM HEPES pH 8.0, 50 mM KCl, 1 mM EDTA, 1 mM DTT overnight at 4 °C. The dialysate was passed through StrepTactin and HisTrap columns, and the cleaved protein collected as the flow-through. This was then concentrated and loaded onto a Superdex 200 Increase 10/300 GL column in 20 mM HEPES pH 7.0, 50 mM KCl, 2.5 mM MgCl₂.

## NMR spectroscopy

All NMR experiments were carried out at 25 °C on Bruker NMR spectrometers performing at 600, 700, 800 and 950 MHz, equipped with cryoprobes. 5 mM MES was included in all NMR samples as an internal pH standard. Experiments were processed using NMRPipe[44] and analysed using CCPNMR Analysis V.2[45]. NMR experiments with full-length Hsc70-T204A were performed in the presence of an ATP-regenerating system, which contained 20 mM creatine phosphate, 2.5 mM ATP, 170 units/mL creatine phosphokinase (Sigma-Aldrich C3755). GF-linker mutations in DNAJB6 (Fig. 4) were performed on a construct of full-length DNAJB6 that lacks the ST domain (ΔST-DNAJB6)[17] but contains CTD. The ΔST-DNAJB6 data are highly similar to that of Abayev-Avraham et al.[23] with no chemical shift changes between the wild-type and the mutants observed past residue 110.

## Backbone and side-chain assignments and collection of nuclear Overhauser effect (NOE) data

Where possible, backbone assignments for DNAJB6 and DNAJB1 constructs were transferred from previously published assignments[17,18,23]. To confirm assignments for mutants with large chemical shift changes and to assign DNAJB6b JD-GF, JD-GSS and JD-GSS-α5 a standard set of HNCACB, CBCACONH, HNCO and HN(CA)CO triple resonance spectra was used. All assignments were carried out in 20 mM sodium

phosphate pH 7.0, 50 mM NaCl, 0.02% NaN₃, 5% D₂O at 600 MHz and 25 °C. 93% (DNAJB1 JD-GF-α5 and its mutants) and 87% (DNAJB6 JD-GF) of the amide resonances of non-proline residues, were successfully assigned. Secondary structure propensities were calculated from the assigned backbone chemical shifts ($^{13}C_\alpha$, $^{13}C_\beta$, $^{13}C'$, $^{15}N$ and $^1H_N$) using TALOS+[46]. Aliphatic and aromatic $^1H$-$^{13}C$ HSQCs and NOESY experiments were performed on samples in 100% D₂O at 700 and 800 MHz. Assignments of aromatic side-chains were derived from a combination of 3D aromatic and aliphatic $^1H$-$^{13}C$ NOE-HMQC and 3D $^1H$-$^{13}C$ HMQC-NOE-HMQC (HCC-NOESY) experiments. All NOESY experiments used a mixing time of 200 ms. Tripe resonance and NOESY experiments were collected with non-uniform sampling and reconstructed with the SMILE[47] plugin for NMRPipe.

## Chemical shift perturbations (CSPs)

CSPs were measured as the difference in the $^1H$ and $^{15}N$ peak positions in $^1H$-$^{15}N$ HSQC spectra and the combined chemical shift perturbation was calculated for each residue using the relationship $\Delta\delta_{comb} = \sqrt{(\Delta\delta_H)^2 + (0.2 \cdot \Delta\delta_N)^2}$, where $\Delta\delta_H$ and $\Delta\delta_N$ were the changes in $^1H$ and $^{15}N$ chemical shifts respectively. A corrected standard deviation to zero, as described in Schumann et al.[48], was used to establish a significance cutoff for CSPs. Briefly, a standard deviation is calculated for all CSPs, and any values in the dataset more than 3 standard deviations over the mean are removed. A new corrected standard deviation is then calculated using the remaining values. This procedure is repeated iteratively until no values in the dataset used to calculate the corrected standard deviation exceed 3 corrected standard deviations above the mean.

## Backbone dynamics

$^{15}N$-$R_{1\rho}$, $^{15}N$-$R_1$ rates and $^{15}N$ {$^1H$} heteronuclear NOE measurements on JDP constructs were recorded at 600 MHz and 25 °C, with sample concentrations specified in the relevant figure legends. The effective spin-lock field for the $^{15}N$-$R_{1\rho}$ experiments was 1.5 kHz. $^{15}N$-$R_2$ values were calculated from $R_{1\rho}$ and $R_1$ using the relationship $R_2 = (R_{1\rho} - R_1\cos^2\theta)/\sin^2\theta$, where $\theta$ is the angle between the effective spin-lock field and the external magnetic ($B_O$) field. The interscan delay for the $^{15}N$-$R_{1\rho}$ and $^{15}N$-$R_1$ experiments was set to 2 s. $^{15}N$ {$^1H$} heteronuclear NOEs were recorded with a saturation period of 4 s followed by a relaxation delay of 1 s. Due to spectral overlap, measurements on DNAJB6 JD-GF were recorded as above but with an HNCO readout. Transverse cross-correlation rates ($\eta_{xy}$) were measured using the pulse sequence of Kroenke et al.[49].

## Residual dipolar coupling (RDC) measurements

RDC measurements were collected at 600 MHz on samples of 200 μM $^{13}C$, $^{15}N$-labelled ΔST-DNAJB6b-F91L in 20 mM sodium phosphate pH 7.4, 100 mM NaCl, aligned in 13 mg/mL bacteriophage Pf1 (ASLA Scientific) (increased NaCl concentration and pH were required to decrease the strength of electrostatic interactions between the protein and alignment medium). Backbone amide $^1D_{NH}$ and $^2D_{C'H}$ RDCs were measured using the ARTSY[50] and TATER[27] HNCO pulse sequences respectively. $^1D_{NH}$ RDCs ranged from -7 to 12 Hz and $^2D_{C'H}$ RDCs ranged from −5 to 4 Hz. RDC data were fitted to an AlphaFold model of DNAJB6 JD-GF-α5 using residues 1–75 and 100–105 but excluding residues 48, 57, 58, 68, 69 using a singular value decomposition approach in XPLOR-NIH[51].

## Hydrogen exchange (HX) NMR measurements

HX rates were measured for DNAJB1 wild-type JD-GF-α5 and its mutants in 20 mM sodium phosphate pH 7.0, 50 mM NaCl at 600 MHz and 25 °C using the WEX-III TROSY pulse sequence[31], with a recovery delay of 4 s and durations of the water inversion interval, *T*, ranging from 0.5

to 900 ms. Log protection factors were calculated as $\log(P) = \log(k_{int}/k_{obs})$, where $k_{int}$ is the intrinsic hydrogen exchange rate for the proton in a disordered, water-accessible random coil conformation[52]. $k_{int}$ rates were multiplied by a scaling factor of 1.5 to prevent negative log(P) values.

## $^{15}$N-Carr-Purcell-Meiboom-Gill (CPMG) relaxation dispersion measurements

$^{15}$N-CPMG relaxation dispersion experiments were recorded at 600, 800 and 950 MHz using a pulse scheme with amide proton decoupling that measures the rates of in-phase $^{15}$N coherences[53]. For Hsc70 binding experiments, Hsc70 was added to a molar ratio of 0.1 and JDP concentrations were as follows: DNAJB6 JD-GF 200 μM, DNAJB6 JD 350 μM, DNAJB1 300 μM. The constant time relaxation delay was set to 40 ms. For the DNAJB6 JD-GF control experiment, 300 μM DNAJB6 JD-GF was used with a constant time relaxation delay of 70 ms. $^{1}$H$_{N}$ constant wave decoupling was applied at a radiofrequency field strength of 10 kHz. The experiment recorded with the relaxation period omitted served as a reference for the calculation of $R_{2,\,eff}$ rates as a function of CPMG field, $\nu_{CPMG}$, as described previously[53]. Uncertainties in $R_{2,\,eff}$ values were obtained from duplicate measurements at two different $\nu_{CPMG}$ frequencies.

## Fitting of NMR relaxation data

$^{15}$N-CPMG and $\delta_{ex}$ data were fitted simultaneously to the kinetic model:

$$\mathrm{JDP} + \mathrm{Hsc} \underset{k_{off}}{\overset{k_{on}}{\rightleftharpoons}} \mathrm{JDPHsc} \tag{1}$$

using in-house scripts written in Python which employed the lmfit module[54]. To allow for data at various total JDP and Hsc concentrations ([JDP]$_{tot}$, [Hsc]$_{tot}$) to be fitted simultaneously, the second order binding nature of the NMR-observable JDP binding to Hsc is considered. In this treatment, the pseudo-first order rate constant $k_{on}^{app}$ that enters the exchange matrix is given by $k_{on}^{app} = k_{on}[\mathrm{Hsc}]_{free}$. The concentration of free Hsc is calculated as $[\mathrm{Hsc}]_{free} = [\mathrm{Hsc}]_{tot} - p_{B}^{JDP}[\mathrm{JDP}]_{tot}$ where $p_{B}^{JDP}$ represents the population of the bound JDP. At each iteration of the optimisation $p_{B}^{JDP}$ is calculated using the quadratic equation obtained from material balance:

$$p_{B}^{JDP} = \left\{ [\mathrm{JDP}]_{tot} + [\mathrm{Hsc}]_{tot} + K_{d} \right.$$
$$\left. - \sqrt{\left( [\mathrm{JDP}]_{tot} + [\mathrm{Hsc}]_{tot} + K_{d} \right)^{2} - 4[\mathrm{JDP}]_{tot}[\mathrm{Hsc}]_{tot}} \right\} / 2[\mathrm{JDP}]_{tot} \tag{2}$$

where $K_{d} = k_{off}/k_{on}$. The exchange rate of the interaction, $k_{ex}$ was calculated as $k_{ex} = k_{on}^{app} + k_{off}$.

The set of the residue-specific optimisation parameters comprised $\left\{ R_{2}^{A}, \Delta\omega_{B} \right\}$ with $\Delta\omega_{B}$ representing the chemical shift difference between the apo and bound states. The $R_{2}$ rate of Hsc70-bound complex $\left( R_{2}^{B} \right)$, was assumed to be five times larger than that of apo JDP $\left( R_{2}^{A} \right)$ based on the molecular weight difference of the species. $\delta_{ex}$ values were calculated as described before[55]. For DNAJB6 JD-GF only $^{15}$N-CPMG were used, while for DNAJB6 JD and DNAJB1 JD-GF $^{15}$N-CPMG and $\delta_{ex}$ data for each construct were fitted together. Convergence of the fits was assessed by a grid search around the best fitted values of the fit parameters.

## Fluorescence polarisation binding experiments

Experiments were carried out on a BMG LABTECH CLARIOstar microplate reader at 25 °C with temperature control. All experiments were carried out using hydrolysis deficient Hsc70 mutant T204A, and

300 nM labelled DNAJB6 JD in 20 mM HEPES pH 7.0, 50 mM KCl, 5 mM MgCl$_{2,}$ 5 mM ATP buffer. Calibration of both focal height and gain for both parallel and perpendicular channels was performed using a well containing fluorescently labelled DNAJB6 JD on its own, with the target polarisation set to 3.5 ×10$^{-2}$. Data were then read for wells with either varying Hsc70-T204A concentration (direct binding experiment) or varying DNAJB1 JD concentration in the presence of 40 μM Hsc70-T204A (competition experiment). Data were exported and processed using Python. Polarisation was calculated using the standard equation:

$$P = \frac{I_{\parallel} - I_{\perp}}{I_{\parallel} + I_{\perp}} \tag{3}$$

Binding curves were fitted using SciPy to a four-point logistic curve equation:

$$y = b + \frac{t - b}{1 + \left( \frac{K_{d}}{x} \right)} \tag{4}$$

where b = bottom asymptote of the curve, t = top asymptote of the curve and $K_{d}$ is derived from the point of inflection assuming a simple 1:1 binding model.

## Small-angle X-ray scattering (SAXS)

SAXS experiments of DNAJB6 constructs were measured in 20 mM sodium phosphate pH 7.0, 50 mM NaCl at 22 °C. The SAXS experiments were performed at the CPHSAXS facility using a BioXolver L (Xenocs) using metal jet source (Excillum), with a wavelength of λ = 1.34 Å, equipped with a Pilatus R 300 K detector (Dectris). Samples were automatically loaded using the BioCUBE sample handling robot from a 96-well tray. Forty to sixty frames (depending on the sample concentration) of 60 s exposures were collected for the protein samples as well as the corresponding buffers. After ensuring the individual frames overlapped, they were averaged, and the background scattering subtracted from the sample scattering. Primary data reduction was made in BIOXTAS RAW[56], consequence background subtraction was performed using PRIMUS from the ATSAS package. Detailed experimental and data analysis parameters are given in Supplementary Table 2.

## Reporting summary

Further information on research design is available in the Nature Portfolio Reporting Summary linked to this article.

## Data availability

NMR chemical shift assignments have been deposited in the Biological Magnetic Resonance Data Bank (BMRB) under the following accession codes: 52736 (DNAJB6 JD-GF), 52762 (DNAJB1 Y92L), and 52763 (DNAJB1 F94L). PDB IDs used in this study include 6U3R (ΔST-DNAJB6) and 6Z5N (DNAJB1 JD-GF-α5). All other data that supports the findings of this study are openly available on figshare (https://doi.org/10.6084/m9.figshare.28163519.v2)[57]. Source data are provided as a Source Data file. Source data are provided with this paper.

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

## Acknowledgements

The work was supported by a Sir Henry Dale Fellowship jointly funded by the Wellcome Trust and the Royal Society (Grant Number 223268/Z/21/Z) to T.K.K. Access to the 700 and 950 MHz spectrometers was provided by the MRC Biomedical NMR Centre at the Francis Crick Institute, which receives core funding from Cancer Research UK (CC1078), the UK Medical Research Council (CC1078), and the Wellcome Trust (CC1078). S.L. and V.F. acknowledge the financial support from the VILLUM FONDEN by the Villum Young Investigator Grants (numbers 19175, 53132), and from the Novo Nordisk Foundation (projects NNF20OC0065260 and NNF22OC0080141). We acknowledge the University of Copenhagen, Small angle X-ray facility, CPHSAXS funded by the Novo Nordisk Foundation (grant no. NNF19OC0055857).

## Author contributions

B.H., F.O., A.P.K. and T.K.K. conceived and designed experiments. B.H., N.L., F.O. and E.K. prepared samples. B.H., N.L., F.O., E.K. and T.K.K. collected and analysed the NMR data. S.L. and V.F collected and analysed the SAXS data. B.H. and T.K.K. wrote the initial draft and all authors have reviewed and edited the manuscript.

## Competing interests

The authors declare no competing interests.
