## [Transparent Peer Review file · Nature Communications]

A low-complexity linker as a driver of intra- and intermolecular interactions in DNAJB chaperones

Corresponding Author: Dr Theodoros Karamanos

Version 0:

Reviewer comments:

Reviewer #1

(Remarks to the Author)

This is a highly interesting and well-conducted study of the role of the low complexity region in DNAJB6b and the role of this region in regulating the interaction with HSP70.

However, the manuscript needs clarification on the following points:

1) In the beginning of the manuscript Hsc70 and Hsp70 are used interchangeably, but not until on page 10 is Hsc70 defined as “constitutively expressed form of human Hsp70”. I suggest this definition is moved to the first mention of Hsc70. Or in case I misunderstood this as a definition, and there is another definition everyone is supposed to know, please correct and clarify.

2) I cannot find SBD is ever being spelled out.

3) In the abstract and introduction, we learn that the study reveals “a remarkable degree of allosteric communication between G-rich, the Fx-repeats and helix 5”. In the results section, “dramatic allostery” is mentioned on line 198 in the sentence “Instead, the Phe to Leu substitution at position 9 causes a dramatic allosteric reorganization in G-rich, with residues 70-74 folding into a stable helix as revealed by their backbone chemical shift.”, and also on lines 237, 280 and 295 in descriptive terms, and then again mentioned in descriptive terms in the discussion section on page 380. Can you give some quantification in terms of free energy difference ($\Delta\Delta G$) or affinity difference (e.g. the factor of affinity reduction) to motivate the use of dramatic and remarkable”?

Allostery is usually defined as a free energy coupling (or affinity enhancement or decrease) for one substrate upon binding of another substrate, or as the shift in a folding-unfolding equilibrium upon binding of a substrate. Thus, the definition of allostery contains a quantitative measure. The allosteric coupling should therefore be reported in quantitative terms, measured or at least estimated, to motivate the use of the wording “remarkable” and “dramatic” allosteric communication”? I assume these terms imply at least 10 kJ/mol in the magnitude of $\Delta\Delta G$, but the cutoff above which it is called remarkable should at least be defined to help the reader with reference to known values of $\Delta\Delta G$ in other allosteric systems.

4) The rate constants k_{on} and k_{off} are clearly defined, as is also true for k_{appon} . But Figure 6 reports k_{ex} , which is not defined. Do you use $k_{ex} = k_{off} + k_{appon}$ or some other definition? Please define in the Figure caption, or in the main text or in Methods section.

In Figure 7 there are reported some rate constant values. It needs to be stated what are these values. Is it k_{ex} or k_{off} ? And if k_{ex} it needs to be defined, as above. In the text it is written that Figure 7 shows faster binding? Do you mean higher k_{on} or higher k_{appon} ? These need to be explicitly calculated from the k_{ex} or k_{off} (whatever is shown) values in Figure 7. Please carry out these calculations and report the values in the main text to help the reader.

5) Several unfortunate typos need correction, a complete language correction is recommended. Here are just some examples:

“slower k_{off} ” -> “lower k_{off} ” or “slower dissociation”

“weaker affinity” -> “lower affinity” or “weaker binding”

“G-rich region” is sometimes called just “G-rich”

“GF-linker” is sometimes called just “GF”

“hypothesise thatseems to induce” -> “hypothesise that induces”

Reviewer #2

(Remarks to the Author)

The manuscript submitted by Karamanos and co-workers is a highly rigorous NMR focused analysis of an intriguing region in the Hsp70 co-chaperone family, DnaJs: the Gly-Phe rich region. There is no question that they have presented data that support their interpretation, and the number of NMR experiments carried out is truly impressive. This actually leads to the main problem with this paper--A non-NMR expert reader will quickly get lost in the experimental material and lose the importance of the authors' model and the general implications they can offer about GF rich domains (or even intrinsically disordered regions). Moreover, one of the most exciting aspects of this study is the difference between two J-proteins, DnaJB1 and DnaJB6, and how subtle a sequence variation between them emerges as such an important functional modulator. Even in the abstract this important finding is relegated to the last sentence.

Here are some suggestions for the authors to consider in a revised manuscript:

1. Bring out the the model and the distinction between the two J proteins earlier and with more emphasis.
2. Gly-Phe rich regions, or more generally Gly-rich, aromatic-rich regions are in many other proteins, including other chaperones like GroEL. The authors should discuss the potential generality of their findings more fully.
3. Put more of the data in supporting information. For example, the RDCs do not seem essential to the interpretation, and certainly the comparison of the two vectors that shows validity in using the RDCs is fodder for the supporting information. The most important conclusion from the RDCs relates to the helical region. Can this be the focus in their use?
4. There are comparisons drawn between DnaJB6 with and without its GF rich domain. The comparison to DnaJB1 is striking (e.g., Fig. 6). But the control of removing the GF rich domain from DnaJB1 was not done. Why?
5. The idea that the GF region acts allosterically is introduced but not well justified.
6. Small suggestion: in text lines 163 to 171 the authors should refer to the relevant figures.

Bottom line: This is tour-de-force application of state-of-the-art NMR methods to deduce the impact of an intrinsically disordered region on the functional properties of a J protein. It should be published, but its impact will be greatly enhanced by making the NMR approaches more accessible to a broad readership and moving non-essential NMR experiments to the SI.

Reviewer #3

(Remarks to the Author)

This manuscript presents a comprehensive investigation into the role of the low-complexity GF-linker in DNAJB chaperones, combining solution NMR and SAXS techniques. The study offers novel insights into how the GF-linker mediates both intra- and intermolecular interactions, contributing to autoinhibition and Hsp70 binding. The findings have significant implications for understanding the regulation of the Hsp70 cycle and related diseases such as limb-girdle muscular dystrophy type 1D (LGMDD1). The work is well-structured and provides compelling data. However, several issues need to be addressed before the manuscript can be considered for publication.

Major concerns :

1. The observation that the GF-linker influences Hsc70 binding kinetics is intriguing, yet the functional consequences of this modulation are unclear. Perform Hsc70 ATPase activity assays in the presence of wild-type and mutant DNAJB constructs to assess how GF-linker dynamics influence chaperone cycling.
2. While the combination of NMR and SAXS techniques is commendable, the rationale behind choosing specific constructs (JD-GF- α 5, JD-GF) needs more justification. Explaining why these constructs are representative would strengthen the methodology section.
3. The manuscript suggests that the GF-linker modulates Hsp70 binding affinity. However, the mechanistic basis behind the observed class-specific differences remains unclear. Additional kinetic analyses could further elucidate these interactions.
4. The authors discuss LGMDD1-associated mutations in DNAJB6 but do not provide functional assays linking structural observations to chaperone activity. Including aggregation suppression assays would strengthen the disease relevance of the findings.

Minor concerns :

1. Ensure consistent use of terms such as "GF-linker," "Fx-repeats," and "autoinhibition" throughout the manuscript.
2. Provide buffer compositions and temperatures used during SAXS measurements in the Methods section.

Version 1:

Reviewer comments:

Reviewer #1

(Remarks to the Author)

The authors have addressed all my comments in a satisfactory manner.

Reviewer #2

(Remarks to the Author)

The authors have responded constructively and thoroughly to my previous critique and as far as I can glean, to those of the other reviewers. The manuscript is now acceptable for publication in my view.

Reviewer #3

(Remarks to the Author)

The authors addressed my questions. I support publication of this work.

Reviewer #1 (Remarks to the Author):

This is a highly interesting and well-conducted study of the role of the low complexity region in DNAJB6b and the role of this region in regulating the interaction with HSP70.

However, the manuscript needs clarification on the following points:

1) In the beginning of the manuscript Hsc70 and Hsp70 are used interchangeably, but not until on page 10 is Hsc70 defined as “constitutively expressed form of human Hsp70”. I suggest this definition is moved to the first mention of Hsc70. Or in case I misunderstood this as a definition, and there is another definition everyone is supposed to know, please correct and clarify.

We thank the reviewer for their positive feedback and for pointing this inconsistency out. We have included a definition for Hsc70 the first paragraph of the introduction (line 43). In general, we use the term Hsp70 to refer to the entire family of proteins and its features, while in the Results we refer to Hsc70, the isoform of Hsp70 which is used in our study.

2) I cannot find SBD is ever being spelled out.

SBD is defined on lines 88 and 297.

3) In the abstract and introduction, we learn that the study reveals “a remarkable degree of allosteric communication between G-rich, the Fx-repeats and helix 5”. In the results section, “dramatic allostery” is mentioned on line 198 in the sentence “Instead, the Phe to Leu substitution at position 9 causes a dramatic allosteric reorganization in G-rich, with residues 70-74 folding into a stable helix as revealed by their backbone chemical shift.”, and also on lines 237, 280 and 295 in descriptive terms, and then again mentioned in descriptive terms in the discussion section on page 380. Can you give some quantification in terms of free energy difference ($\Delta\Delta G$) or affinity difference (e.g. the factor of affinity reduction) to motivate the use of dramatic and remarkable”?

Allostery is usually defined as a free energy coupling (or affinity enhancement or decrease) for one substrate upon binding of another substrate, or as the shift in a folding-unfolding

equilibrium upon binding of a substrate. Thus, the definition of allostery contains a quantitative measure. The allosteric coupling should therefore be reported in quantitative terms, measured or at least estimated, to motivate the use of the wording “remarkable” and “dramatic” allosteric communication”? I assume these terms imply at least 10 kJ/mol in the magnitude of $\Delta\Delta G$, but the cutoff above which it is called remarkable should at least be defined to help the reader with reference to known values of $\Delta\Delta G$ in other allosteric systems.

Indeed, allostery is associated with changes in binding affinity for one substrate upon binding of another one as the reviewer points out. However, allostery can also be more broadly defined as changes in one part of a molecule in response to alterations occurring at a distinct more distant site. This is exactly what we observe using the panel of GF mutations in Figures 4 and 5. To answer the reviewers point we have removed instances of ‘dramatic’ and ‘remarkable’ in relation to the allosteric changes observed and have substituted allostery with long-range interactions where appropriate to avoid confusion.

4) The rate constants k_{on} and k_{off} are clearly defined, as is also true for k_{appon} . But Figure 6 reports k_{ex} , which is not defined. Do you use $k_{ex} = k_{off} + k_{appon}$ or some other definition? Please define in the Figure caption, or in the main text or in Methods section. In Figure 7 there are reported some rate constant values. It needs to be stated what are these values. Is it k_{ex} or k_{off} ? And if k_{ex} it needs to be defined, as above. In the text it is written that Figure 7 shows faster binding? Do you mean higher k_{on} or higher k_{appon} ? These need to be explicitly calculated from the k_{ex} or k_{off} (whatever is shown) values in Figure 7. Please carry out these calculations and report the values in the main text to help the reader.

We thank the reviewer for this useful comment. The definition of k_{ex} which is indeed equal to $k_{on}^{app} + k_{off}$ has been added to the legend of Figure 6 and in the Methods section, line 625. The rate constant values in Figure 7 refer to k_{ex} and this has been included in the figure legend, along with its definition. To make things even more clear a new Supplementary Table 1 is included that shows all the fitted kinetic parameters from the CPMG data. As it is hopefully now clear from Supplementary Table 1, the lifetime of the complex is dictated by k_{off} .

5) Several unfortunate typos need correction, a complete language correction is recommended. Here are just some examples:

“slower koff” -> “lower koff” or “slower dissociation”

“weaker affinity” -> “lower affinity” or “weaker binding”

“G-rich region” is sometimes called just “G-rich”

“GF-linker” is sometimes called just “GF”

“hypothesise thatseems to induce” -> “hypothesise that induces”

We have gone through the manuscript and corrected all identified typographical errors.

Reviewer #2 (Remarks to the Author):

The manuscript submitted by Karamanos and co-workers is a highly rigorous NMR focused analysis of an intriguing region in the Hsp70 co-chaperone family, DnaJs: the Gly-Phe rich region. There is no question that they have presented data that support their interpretation, and the number of NMR experiments carried out is truly impressive. This actually leads to the main problem with this paper--A non-NMR expert reader will quickly get lost in the experimental material and lose the importance of the authors' model and the general implications they can offer about GF rich domains (or even intrinsically disordered regions). Moreover, one of the most exciting aspects of this study is the difference between two J-proteins, DnaJB1 and DnaJB6, and how subtle a sequence variation between them emerges as such an important functional modulator. Even in the abstract this important finding is relegated to the last sentence.

Here are some suggestions for the authors to consider in a revised manuscript:

1. Bring out the model and the distinction between the two J proteins earlier and with more emphasis.

We thank the reviewer for this suggestion and appreciate that some of the important conclusions of this work could be emphasised earlier in the manuscript. The abstract and final paragraph of the introduction have been rewritten to better highlight the striking differences in structural behaviour of the GF-linker between DNAJB1 and DNAJB6.

2. Gly-Phe rich regions, or more generally Gly-rich, aromatic-rich regions are in many other proteins, including other chaperones like GroEL. The authors should discuss the potential generality of their findings more fully.

We agree with the reviewer that this important aspect was lacking in our original submission. We have now expanded the last paragraph of the Discussion (lines 466 - 478) to discuss the importance of Gly-/Aromatic-rich regions in proteins other than DNAJs. We hope that this now better reflects the general implications of our findings to other systems.

3. Put more of the data in supporting information. For example, the RDCs do not seem essential to the interpretation, and certainly the comparison of the two vectors that shows validity in using the RDCs is fodder for the supporting information. The most important conclusion from the RDCs relates to the helical region. Can this be the focus in their use?

We have edited the RDC section to make it more readable to a non-NMR expert. The comparison between the two alignment media is now moved to the legend of Supplementary Figure 5 as suggested. We opted to keep the RDC analysis in the main text as we believe that data serve two essential purposes for our story: (1) confirming that helix 5 remains docked in the F91L mutant and (2) demonstrating that the newly formed helix L adopts a DNAJB1-like conformation. Both findings are critical for understanding the structural consequences of the Phe to Leu mutations in DNJAB6 and highlight their long-range effects. Given the short length of helix L, both RDC datasets are necessary to validate its orientation and thus, removing them from the main text would weaken the structural interpretation of for the structural reorganisation of the GF-linker.

4. There are comparisons drawn between DnaJB6 with and without its GF rich domain. The comparison to DnaJB1 is striking (e.g., Fig. 6). But the control of removing the GF rich domain from DnaJB1 was not done. Why?

We have now performed CPMG experiments for the association of DNAJB1 JD with Hsc70. Our preliminary $k_{\text{ex}} \sim 2000 \text{ s}^{-1}$ and $K_{\text{d}} \sim 450 \text{ }\mu\text{M}$ values are in agreement with our prior data shown in Supplementary Figure 14 which clearly suggest a lower affinity of DNAJB1 JD for Hsc70 in comparison to DNAJB6 JD. Strikingly, they are also similar to those shown in Figure 6C for DNAJB1 JD-GF ($\sim 1790 \text{ s}^{-1}$ and $\sim 288 \text{ }\mu\text{M}$ respectively). This result suggests even further

that the highly flexible GF-linker of DNAJB1 does not participate in Hsc70 binding in the open JD-GF. A complete analysis of this data however is a significant task that will require more experiments at additional magnetic fields in order to finalise the fits and get reliable error estimates. Therefore, and since our manuscript is already NMR-data heavy, we decided to not include the full DNAJB1 JD CPMG data which we will present in a subsequent publication. We have however included the preliminary data discussed here in Supplementary Table 1 as we do not expect these values to change significantly during further analysis.

5. The idea that the GF region acts allosterically is introduced but not well justified.

Please refer to the response to point 3 from reviewer 1. We have substituted allostery with long-range interactions where appropriate to reflect the effect of aromatic substitutions in the GF-linker that extend to more than 20 residues N- (DNAJB6) or C-terminally (DNAJB1).

6. Small suggestion: in text lines 163 to 171 the authors should refer to the relevant figures.

References have been added to the appropriate figures.

Bottom line: This is tour-de-force application of state-of-the-art NMR methods to deduce the impact of an intrinsically disordered region on the functional properties of a J protein. It should be published, but its impact will be greatly enhanced by making the NMR approaches more accessible to a broad readership and moving non-essential NMR experiments to the SI.

We thank the referee for their positive comments about our work.

Reviewer #3 (Remarks to the Author):

This manuscript presents a comprehensive investigation into the role of the low-complexity GF-linker in DNAJB chaperones, combining solution NMR and SAXS techniques. The study offers novel insights into how the GF-linker mediates both intra- and intermolecular interactions, contributing to autoinhibition and Hsp70 binding. The findings have significant implications for understanding the regulation of the Hsp70 cycle and related diseases such as limb-girdle muscular dystrophy type 1D (LGMDD1). The work is well-structured and provides

compelling data. However, several issues need to be addressed before the manuscript can be considered for publication.

Major concerns:

1. The observation that the GF-linker influences Hsc70 binding kinetics is intriguing, yet the functional consequences of this modulation are unclear. Perform Hsc70 ATPase activity assays in the presence of wild-type and mutant DNAJB constructs to assess how GF-linker dynamics influence chaperone cycling.

To answer this point, we refer the reviewer and the readers to previous elegant work by the Rosenzweig group (references 18 and 23) which has performed ATP activity assays on LGMDD1 DNAJB6 mutants, including F91L and F93L. This work establishes a correlation between these mutations and an increased ability of DNAJBs to enhance the ATPase activity of Hsc70. Our data now gives a direct mechanistic interpretation for these prior observations. We have added a sentence in line 290 to specifically refer to these data.

2. While the combination of NMR and SAXS techniques is commendable, the rationale behind choosing specific constructs (JD-GF- α 5, JD-GF) needs more justification. Explaining why these constructs are representative would strengthen the methodology section.

We thank the reviewer for their comment. We have changed the first section of our Methods to include the design of the JDP constructs, which now provides a clear rationale for choosing these specific truncations.

3. The manuscript suggests that the GF-linker modulates Hsp70 binding affinity. However, the mechanistic basis behind the observed class-specific differences remains unclear. Additional kinetic analyses could further elucidate these interactions.

Indeed, and as pointed out by reviewer 2 the class-dependent affinity for Hsc70 is one of the important findings of our study. In Figures 2 and Supplementary Figure 2 we show that the GF-linkers of DNAJB1 and DNAJB6 in their 'open' states show very different conformational properties with the one being intrinsically disordered (DNAJB1) and the other showing residual secondary and tertiary structure (DNAJB6). Mechanistically, their different conformational

properties are behind the different effects of the two GF-linkers on the binding affinity/kinetics for Hsc70. The disordered GF-linker of the open DNAJB1 JD-GF cannot bind the SDB β while the partially collapsed GF-linker of DNAJB6 JD-GF can. Indeed, restricting the conformational flexibility of DNAJB1 GF-linker by introducing the ‘closed’ JD-GF- α 5 state allows it to be recognised by the SDB β (Figure 6D). As the referee suggests, we have performed additional CPMG experiments on the free JD of DNAJB1 (see response to point 4 by reviewer 2). These new results show that the affinity and kinetics of Hsc70 binding are not influenced by the GF-linker of DNAJB1, strongly contrasting with the results obtained for the equivalent constructs in DNAJB6. We have amended lines 458-462 in the Discussion and included a new Supplementary Table 1 to make the mechanistic basis of class-specific Hsc70 recognition clearer.

4. The authors discuss LGMDD1-associated mutations in DNAJB6 but do not provide functional assays linking structural observations to chaperone activity. Including aggregation suppression assays would strengthen the disease relevance of the findings.

We feel that this valid comment is outside the scope of our study. In our manuscript we use LGMDD1 mutations primarily to investigate the conformational properties of the GF-linker and not to elucidate disease mechanisms. Work from Abayev-Avraham et al (Ref 23) has used a much more extensive panel of LGMDD1 mutations to understand how they are related with disease. Using aggregation suppression assays they show that none of the LGMDD1 mutations have any effect on the Hsp70-independent antiaggregation activity of DNAJB6. Instead, they propose that the mutations increase the population of the ‘open’/uninhibited JDP state leading to unregulated Hsp70 binding and eventually disease. In our study, we saw no evidence of an increased population of the ‘open’ JD state for two of the most severe LGMDD1 mutants (Figure 4 and Supplementary Figure 4). However, we show that the Fx repeats region, where the most severe LGMDD1 mutations localise, is directly involved in Hsc70 binding, a fact that could lead to unregulated Hsp70 binding as observed by Abayev-Avraham *et al.* We have amended the Discussion (lines 447 - 450) to better reflect these ideas.

Minor concerns :

1. Ensure consistent use of terms such as "GF-linker," "Fx-repeats," and "autoinhibition" throughout the manuscript.

Changes have been made to ensure more consistent use of terminology.

2. Provide buffer compositions and temperatures used during SAXS measurements in the Methods section.

The buffer composition for the SAXS experiments (20 mM sodium phosphate pH 7.0, 50 mM NaCl) is included in the SAXS methods section, and the temperature (22°C) has been added.